# MAPK pathway mutations in head and neck cancer affect immune microenvironments and ErbB3 signaling

Hoi-Lam Ngan[1],*, Yuchen Liu[1],*, Andrew Yuon Fong[2], Peony Hiu Yan Poon[1], Chun Kit Yeung[1], Sharon Suet Man Chan[3], Alexandria Lau[1], Wenying Piao[1], Hui Li[1], Jessie Sze Wing Tse[1], Kwok-Wai Lo[4], Sze Man Chan[1], Yu-Xiong Su[5], Jason Ying Kuen Chan[6], Chin Wang Lau[7], Gordon B Mills[8], Jennifer Rubin Grandis[9], Vivian Wai Yan Lui[1]

**MAPK pathway mutations affect one-fifth of head and neck squamous cell carcinoma (HNSCC). Unexpectedly, MAPK pathway aberrations are associated with remarkably long patient survival, even among patients with *TP53* mutations (median ~14 yr). We explored underlying outcome-favoring mechanisms with omics followed by preclinical models. Strikingly, multiple hotspot and non-hotspot MAPK mutations (*A/BRAF*, *HRAS*, *MAPK1*, and *MAP2K1/2*) all abrogated ErbB3 activation, a well-established HNSCC progression signal. Inhibitor studies functionally defined ERK activity negatively regulating phospho-ErbB3 in MAPK-mutants. Furthermore, pan-pathway immunoprofiling investigations identified MAPK-mutant tumors as the only "CD8+ T-cell–inflamed" tumors inherently bearing high-immunoreactive, constitutive cytolytic tumor microenvironments. Immunocompetent MAPK-mutant HNSCC models displayed active cell death and massive CD8+ T-cell recruitment in situ. Consistent with CD8+ T-inflamed phenotypes, MAPK-mutant HNSCC patients, independent of tumor-mutational burden, survived 3.3–4 times longer than WT patients with anti-PD1/PD-L1 immunotherapies. Similar prognosticity was noted in pan-cancers. We uncovered clinical, signaling, and immunological uniqueness of MAPK-mutant HNSCC with potential biomarker utilities predicting favorable patient survival.**

## Introduction

The incidence of head and neck squamous cell carcinoma (HNSCC) is approaching 0.7 million/year globally (Vigneswaran & Williams, 2014). Despite extensive genomic and molecular characterizations of HNSCC primary tumors, the clinical significance of somatic aberrations in HNSCC remains underexplored. As of today, we still lack clinically relevant genomic biomarkers for effective management of HNSCC patients. This is largely due to our lack of understanding of the precise tumor biology and potential clinical utility of the genomic signatures of patients' tumors.

For HNSCC, recent genomic studies have revealed that human papillomavirus (HPV) positivity in patients' tumors can be potentially used as a favorable prognostic biomarker (mainly for cancer from the oropharynx subsite), which can clinically stratify patients for treatment de-intensification purposes (Gillison et al, 2019; Mehanna et al, 2019; Pearlstein et al, 2019). The biological reasons underlying HPV's positive prognosticity have been unfolded and found to be linked to the antigenic nature of HPV and the tumor's *TP53* wild-type (WT) status which allows proper execution of drug-induced apoptosis by functional p53 (Fakhry & Gillison, 2006). Similarly, genomic studies showed that somatic *TP53* mutations could robustly predict poor HNSCC patient outcome, for reasons that mutant p53 could cause drug resistance and radiation resistance due to biological impairment of cancer cell apoptosis in HNSCC. However, the high frequency of *TP53* mutations in >80–85% of primary HNSCC greatly limits their development into useful stratification biomarkers for treatment selection, especially because *TP53*-mutant HNSCC cannot be effectively drugged. Recently, *PIK3CA* mutations have been shown to be predictive of PI3K inhibitor and nonsteroidal anti-inflammatory drug (NSAID) responses in HNSCC, with proven biology demonstrated in PI3K-mutant, PI3K-activated preclinical models of HNSCC and retrospective patient cohorts (Lui et al, 2013; Hedberg et al, 2019). These studies identified drug sensitivity characteristics of PI3K-addicted tumors in HNSCC. Yet, clinical incorporation of *PIK3CA* mutations as candidate-predictive biomarkers for clinical utility still awaits further

[1]School of Biomedical Sciences, Faculty of Medicine, The Chinese University of Hong Kong, Hong Kong, Hong Kong SAR [2]Department of Biochemistry, Case Western Reserve University, Cleveland, OH, USA [3]Faculty of Medicine, Imperial College London, London, UK [4]Department of Anatomical and Cellular Pathology, Faculty of Medicine, The Chinese University of Hong Kong, Hong Kong, Hong Kong SAR [5]Department of Oral and Maxillofacial Surgery, Faculty of Dentistry, The University of Hong Kong, Hong Kong, Hong Kong SAR [6]Department of Otorhinolaryngology, Head and Neck Surgery, Faculty of Medicine, The Chinese University of Hong Kong, Hong Kong, Hong Kong SAR [7]Department of Otorhinolaryngology Head and Neck, Yan Chai Hospital, Hong Kong, Hong Kong SAR [8]Department of Cell, Development and Cancer Biology, Knight Cancer Institute, Oregon Health and Sciences University, Portland, OR, USA [9]Department of Otolaryngology–Head and Neck Surgery, University of California, San Francisco, San Francisco, CA, USA

Correspondence: vlui002@cuhk.edu.hk
*Hoi-Lam Ngan and Yuchen Liu contributed equally to this work

prospective validation in clinical trials. These recent findings demonstrate that a deeper understanding of the clinical impacts of HNSCC genetic aberrations in relation to their underlying biology can potentially reveal new approaches for clinical management of HNSCC.

Here, we first reported that MAPK pathway mutations in HNSCC predict remarkably long patient survival, even among patients bearing *TP53* mutations (median ~14 yr), much longer than that of HPV-positive HNSCC (median ~5.5 yr). The favorable prognosticity of MAPK pathway mutations in HNSCC was found to be independent of HPV. Subsequent molecular dissections revealed two plausible underlying mechanisms operative by MAPK mutations in patient tumors, followed by preclinical HNSCC models. First, multiple hotspot and non-hotspot MAPK mutations (*A/BRAF*, *HRAS*, *MAPK1*, and *MAP2K1/2*) all abrogated ErbB3 activation, an established signal for HNSCC progression. Inhibitor studies functionally defined ERK activity negatively regulating p-ErbB3 in MAPK-mutants. Second, comprehensive pan-pathway immune landscape investigations identified MAPK-mutant tumors as the only "CD8+ T-cell–inflamed" tumors with inherently immunoreactive tumor microenvironments with constitutive cytolysis, subsequently validated in immunocompetent HNSCC models. As low tumoral phospho-ErbB3 levels and elevated CD8+ T-cell infiltrations are recently established events driving good patient survivals in HNSCC (Takikita et al, 2011; de Ruiter et al, 2017), our study first defined somatic MAPK pathway mutations as novel genomic events governing both important outcome-favoring features in HNSCC. Consistent with the identified CD8+ T-cell-inflamed nature, MAPK pathway mutations alone, independent of tumor mutational burden (TMB), were able to identify subsets of HNSCC patients benefiting significantly from anti-PD1/PD-L1 immunotherapy clinically. Similar prognosticity of MAPK pathway mutations were noted in pan-cancer immunotherapy settings. Our study uncovers novel clinical, biological, and immunological uniqueness of MAPK-mutant HNSCC and may indicate important clinical utilities of MAPK pathway mutations in this cancer.

## Results

### MAPK (ERK) pathway-mutated HNSCC patients have remarkably long survivals in two independent cohorts

As of today, the clinical significance of somatic genomic aberrations in HNSCC remains underexplored. Here, using a comprehensive pan-pathway approach, we examined the prognostic impact of mutational events of seven major cancer-related signaling pathways on HNSCC patient survivals with The Cancer Genome Atlas (TCGA) dataset (N = 508, exome and clinical data, June 2019). These include three major mitogenic pathways, the PI3K, MAPK(ERK), and JAK/STAT pathways; two differentiation-related pathways, the Notch and TGF-*β*/Smad pathways; the inflammation-related NF-*κ*B pathway; and the stem cell–related WNT pathway, representing a total of 81% of cases (Fig 1A and B). Pathway members are defined in the Materials and Methods section. About 17% (84/502) of patients are HPV-positive, whereas the remaining 19% lack these pathway aberrations or are HPV-negative/unknown.

Among all seven pathways examined, only mutations in the MAPK pathway (denoted in red for favorable survival) and the Notch

pathway (blue for unfavorable survival) are associated with patient outcomes (Figs 1B and S1A–F). Mutational burdens of all pathway-altered cases are similar (P = n.s.; Fig S1G). Our finding that Notch pathway-mutated HNSCC patients have worsened outcomes are consistent with a recent Taiwanese report that *NOTCH1* mutations predict recurrences with poor outcomes (Liu et al, 2016). Unexpectedly, MAPK pathway mutations, comprising mainly activating hotspot mutations (e.g., *HRAS* p.G12S and *MAPK1* p.E322K [Stransky et al, 2011; Van Allen et al, 2015]; Fig S2), are associated with a doubling of overall survival (OS) with a median of 95.27 versus 47.93 mo for MAPK-WT patients (log-rank test, *P* = 0.0201; Fig 1C). These patients also have a reduced risk of death versus WT patients (OR = 0.5466, *P* = 0.0156, Fisher's exact test).

Most strikingly, MAPK pathway mutations are also prognostic for markedly improved survival even among *TP53*-mutated patients (given *TP53* mutations are usually indicators for HNSCC disease progression and disease aggressiveness) (Fig 1D). In fact, "MAPK and *TP53* double mutant" patients (N = 363) have an extreme long median OS of 169.25 mo (~14 yr), which is 4.77 times longer than that of the "MAPK-WT/*TP53*-mutated" counterparts (35.45 mo; *P* = 0.0074). The double mutant patients also have a 55.26% reduction in chances of death (OR = 0.4474) versus MAPK-WT counterparts (*P* = 0.0063, Fisher's exact test).

Clinically, MAPK pathway mutations are not associated with HPV status, nor clinical staging (P = n.s.), but potentially associated with lower alcohol intake per TCGA alcohol history, and a higher occurrence in females (*P* = 0.01003, 0.03372, respectively, Table S1). Importantly, unlike HPV-positive HNSCC with favorable outcomes, MAPK pathway mutations span multiple head and neck anatomic subsites, including the oral cavity sites, larynx, oropharynx, and others (Table S2). More than 87% (83/95 cases) of MAPK pathway-mutated tumors are HPV-negative. Upon HPV stratification, MAPK pathway mutations are still found to be prognostic for HPV-negative HNSCC (*P* = 0.0352, Fig S3). Interestingly, overexpressions of the MAPK proteome components are also prognostically associated with improved patient survival in HNSCC and 10 additional cancer types (Fig 1E and Table S3), suggesting that MAPK pathway activity may influence patient outcome in HNSCC, and likely in other cancers.

HNSCC patients in TCGA dataset were treated with conventional therapies. Strikingly, in an independent MSK-IMPACT cohort with heavily pretreated advanced and metastatic HNSCC, we cross-validated that MAPK pathway mutations (with nine MAPK pathway members sequenced therein: *H/N/K-RAS*, *A/B-RAF*, *RAF-1*, *MAP2K1/2*, and *MAPK1*) were also prognostic of favorable HNSCC patient outcome (Fig 1F). Although only "targeted therapies" were generally described for this MSK-IMPACT study (Zehir et al, 2017), the apparent cross-cohort, cross-treatment prognostic nature of MAPK mutations seem to suggest inherent outcome-favoring features of these tumors contributive to improved patient survival.

Interestingly, MAPK pathway mutations also predict favorable outcomes in TCGA uterine corpus endometrial carcinoma (Fig 1G), indicating that the positive prognosticity of MAPK pathway aberrations is not restricted to HNSCC.

### Multiple MAPK hotspot and non-hotspot mutations all inhibit ErbB3 phosphorylation in HNSCC

ErbB3 activation contributes to HNSCC proliferation (O-charoenrat et al, 2002; Zhang et al, 2014) and is a key mediator of HNSCC

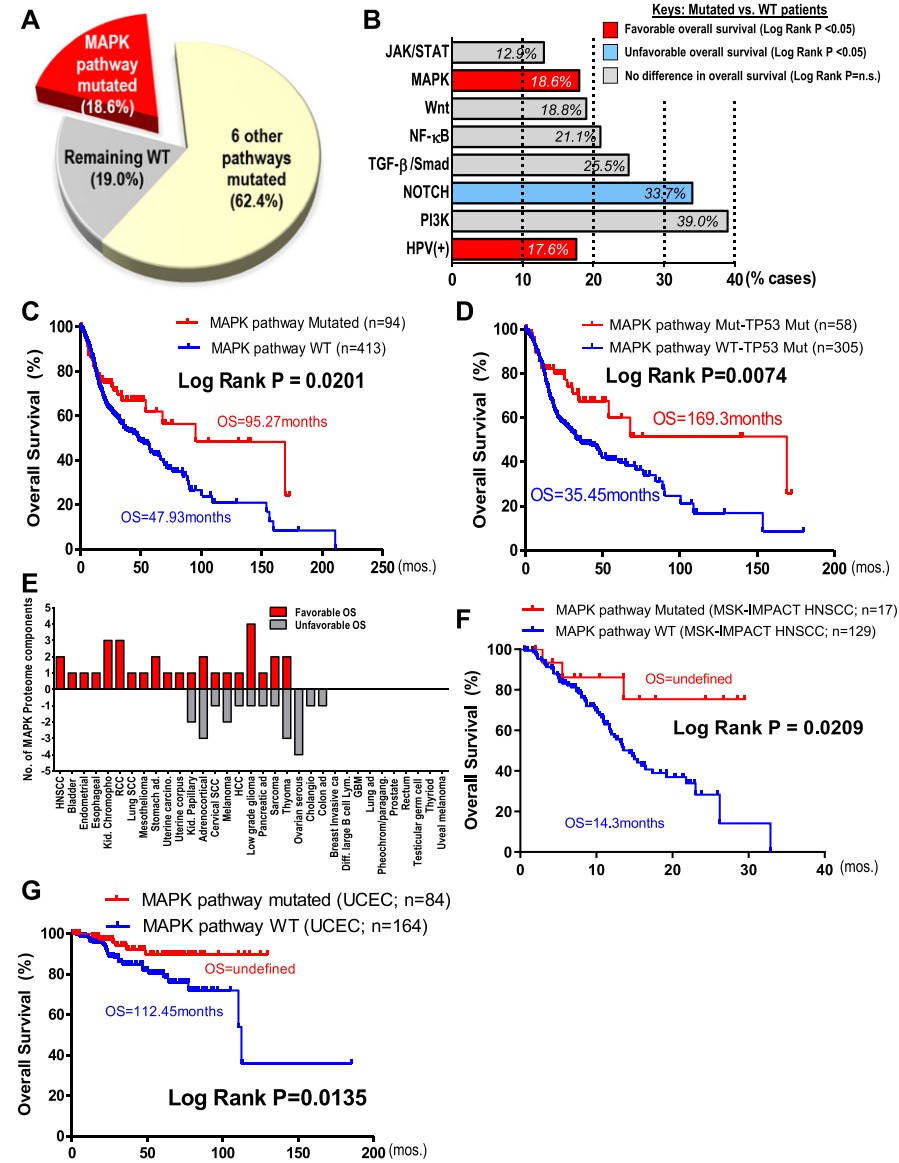

**Figure 1. MAPK pathway mutations in head and neck squamous cell carcinoma (HNSCC) are associated with remarkable patient survival.**
**(A)** Percentage of patients affected by MAPK pathway mutations and mutations of six other cancer pathways (PI3K, JAK/STAT, Notch, WNT, NF-κB, and TGFβ/Smad) in TCGA HNSCC provisional cohort (N = 510). **(B)** HNSCC patient outcome associations for all seven pathway mutations (i.e., pathway-mutated versus pathway WT) and HPV status. Red bars indicate favorable HNSCC overall survival (OS), whereas the blue bar indicates unfavorable OS when the pathway components are mutated (log-rank test *P*-values are shown). **(C)** Kaplan–Meier OS curves for MAPK pathway-mutated HNSCC patients versus MAPK pathway WT patients (TCGA HNSCC provisional cohort). **(D)** Kaplan–Meier OS curves for *TP53*-mutated patients with MAPK pathway-mutated versus WT HNSCC (TCGA HNSCC provisional cohort). **(E)** Bar graph showing the number of MAPK pathway protein components for each cancer type (total 33 cancer types) that were significantly correlated with OS. Red bars indicate associations with favorable outcomes, whereas grey bars indicate associations with unfavorable outcomes in each cancer type, when the MAPK protein component(s) is/are overexpressed (median cutoff; The Cancer Protein Atlas database). **(F)** Kaplan–Meier OS curves for MAPK pathway-mutated HNSCC patients versus MAPK pathway WT patients (MSK-IMPACT HNSCC cohort). **(G)** Kaplan–Meier survival curves showing increased OS for uterine corpus endometrial carcinoma patients with MAPK pathway mutation versus WT (TCGA uterine corpus endometrial carcinoma cohort).

progression. Data from The Cancer Protein Atlas (TCPA, https://tcpaportal.org/tcpa/, (Li et al, 2013, 2017a)) have established tumoral overexpression of phospho-ErbB3(Y1289) as the top 1 protein-signaling event most significantly associated with decreased HNSCC patient survival among 273 key signaling proteins examined (*P* = 0.0006, Fig 2A and Table S4 shows top 20 signaling proteins of significant survival correlations). Overexpressions are defined as levels above median in TCPA. Based on our recent findings that *MAPK1* mutation can alter ErbB family signaling in HNSCC (Van Allen et al, 2015), we thus examined the effects of various major hotspot MAPK pathway mutations on p-ErbB3(Y1289) expression. Strikingly, we found that introduction of *BRAF* p.V600E hotspot mutation in HNSCC cell model, FaDu, strongly inhibited p-ErbB3(Y1289) expression versus control (~77.8% inhibition, *P* = 0.0038, Fig 2B). Furthermore, many other major hotspot mutations, including *HRAS* p.G12V, *MAPK1* p.E322K/p.D321N, *MAP2K1* p.K57N, and *MAP2K2* p.F57L

mutations all inhibited p-ErbB3(Y1289) expressions in HNSCC cell models (Figs 2B and S4A). For *ARAF*, both *ARAF* p.P508L (non-hotspot and HNSCC associated) and the hotspot *ARAF* p.S214F mutations inhibited p-ErbB3(Y1289) expressions. In addition to these mutants, overexpression of either *MAP2K1*-WT or *BRAF*-WT was also sufficient to inhibit p-ErbB3(Y1289) expressions by ~60% and 50%, respectively.

## Mutant-specific ErbB3 inhibition reversed by GDC-0994

To this end, MAPK-activating mutations function to inhibit ErbB3 activation in isogenic HNSCC models, as assessed by p-ErbB3(Y1289) levels. Therefore, we further tested this non-canonical MAPK/ErbB3 relationship using the pharmacological GDC-0994 MAPK1/2-RSK inhibitor in HNSCC cells harboring endogenous MAPK pathway mutations. As shown in Figs 2C and S4B, only MAPK-mutant HNSCC cells showed increases in p-ErbB3(Y1289) upon inhibitor treatment

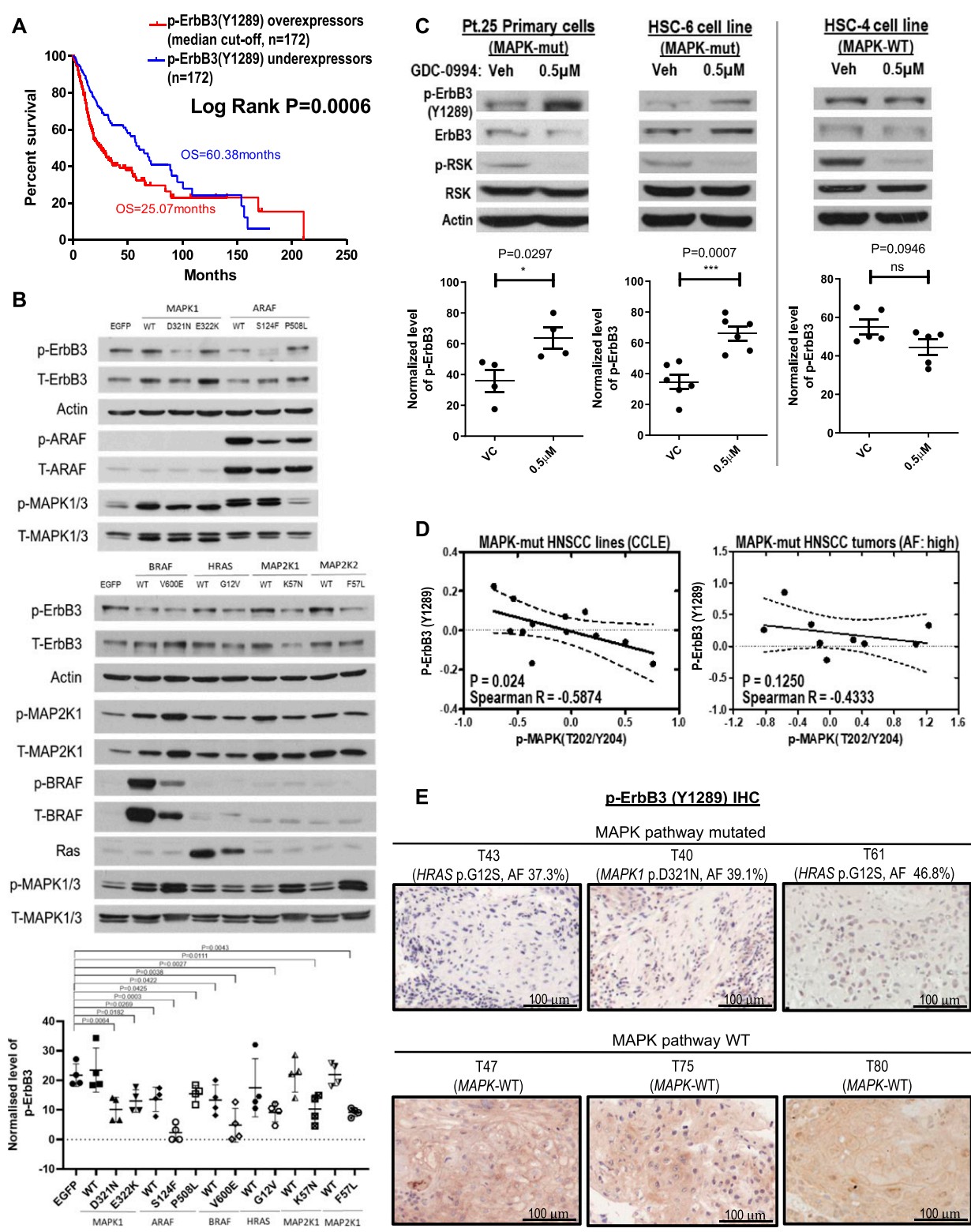

**Figure 2. Multiple MAPK pathway mutations inhibited ErbB3 activation.**
**(A)** Kaplan–Meier overall survival curves for head and neck squamous cell carcinoma (HNSCC) patients, whose tumors overexpressed versus underexpressed phospho-ErbB3(Y1289) (median cutoff) in TCPA HNSCC cohort (N = 344). **(B)** Western blot results of phospho-ErbB3(Y1289) protein levels upon ectopic expression of the *MAPK1*, *ARAF*, *BRAF*, *HRAS*, and *MAP2K1* as well as *MAP2K2* wild-type and mutant constructs in FaDu cells by retroviral infection (pool of at least four independent repeats). **(C)** Western blot results of phospho-ErbB3(Y1289) levels of HNSCC Pt.25 primary tumor cultures (carrying both *HRAS* p.G12S and *MAPK1* p.R135K mutations), HSC-6 cell line (carrying *MAPK1* p.E322K mutation), and HSC-4 (MAPK pathway WT, Cancer Cell Line Encyclopedia [CCLE]) upon MAPK inhibitor GDC-0994 treatment for 30 min. 50 μg of protein lysate

versus vehicle. In a primary patient culture, Pt-25 harboring *MAPK1* p.R135K and *HRAS* p.G12S mutations and the HSC-6 cell line harboring *MAPK1* p.E322K, >100–200% increases of p-ErbB3 expression were observed, whereas such an increase was not observed in MAPK-WT cells (HSC-4). The data suggested MAPK (ERK) activity negatively regulates p-ErbB3 in a mutant-specific manner in HNSCC. Such a mutant-specific phenomenon was further supported by proteomic findings in MAPK-mutant HNSCC cell lines (N = 12, Cancer Cell Line Encyclopedia [CCLE] [Ghandi et al, 2019]) and in relatively homogeneous MAPK-mutant patient tumors with high allele frequencies (>40%), in which p-MAPK(T202/Y204) and p-ErbB3 levels were negatively correlated with high Spearman R correlation coefficients of −0.5874 (*P* = 0.0244) and 0.4333 (*P* = 0.1250, a trend in non-microdissected TCPA tumors; Fig 2D), respectively. In MAPK-WT HNSCC tumors or cell lines, p-MAPK(T202/Y204) and p-ErbB3 were not correlated (P = n.s.; Fig S5). Thus, it is likely that when activating MAPK mutations are present at high allele frequencies in HNSCC tumors, they could substantially inhibit p-ErbB3 expressions in HNSCC. Indeed, immunohistochemistry showed three out of three MAPK-mutant HNSCC patient tumors with high allele frequencies (~40%) all expressed low tumorous p-ErbB3(Y1289), whereas WT tumors show high membranous p-ErbB3(Y1289) staining (Fig 2E). Therefore, MAPK pathway mutations in HNSCC function to inhibit ErbB3 signaling, an established critical signaling dictating poor HNSCC patient survival.

### Treatment-naive MAPK-mutant HNSCC tumors have remarkably active cytolytic immune landscapes

Response to therapy is an important factor determining cancer patient survival. Yet, there was no clear association between MAPK mutations and responses to cisplatin, docetaxel, 5-flurouracil, methotrexate, and cetuximab in CCLE HNSCC drug-sensitivity database (Fig S6A). Similarly, expression of MAPK pathway-mutants in HNSCC cells did not significantly alter chemosensitivity (Fig S6B).

In addition to our proteomics findings, we further examine, by transcriptomic analysis, if MAPK-mutant HNSCC tumors harbor immune features favoring survival as recent findings in melanoma showed that patients with MAPK pathway mutations have remarkable clinical outcome likely due to increased neo-antigenicity or antitumor immune microenvironment (Cadley et al, 2018; Veatch et al, 2018). We compared the transcriptional profiles of MAPK pathway-mutated and WT HNSCC tumors (all treatment-naïve tumors per TCGA standard). As shown in Fig 3A and Supplemental Data 1, there were 1,793 significant differentially expressed genes (DEGs) in MAPK-mutated versus WT HNSCC tumors with *P*-values < 0.05 and false discovery rate (FDR) < 0.05, among which 130 protein-coding DEGs showed a >1.41-fold difference ($log_2$ fold-change > 0.5) in gene expression with FDR < 0.05 (Fig 3B). Strikingly, more than 70% (91/130) of these genes are immune related (Table S5). Gene set enrichment

analysis (GSEA) revealed top four biological processes significantly enhanced in MAPK-mutated tumors as all immune-related (Figs 3C and S7). *Perforin* (*PRF1*), and four granzymes, *GZMA* (*Cytotoxic T-Lymphocyte-Associated Serine Esterase-3*), *GZMB* (*Cytotoxic T-Lymphocyte-Associated Serine Esterase-1*), *GZMH* (*Cytotoxic Serine Protease C*), and *GZMK* (*Tryptase II*) were all significantly up-regulated in MAPK-mutated tumors (Fig 3D), indicating remarkably active, likely cytolytic immune responses in situ. Similar findings were consistently noted in HPV-negative MAPK-mutated tumors (Fig S8A).

### The only tumors with inherently CD8[+] T-cell–inflamed immunoactive, cytolytic tumor microenvironments among seven pathway mutants

Subsequent immune landscape examination across all seven signaling pathways by Tumor Immune Estimation Resource (TIMER) analysis (Li et al, 2016, 2017) reveals for the first time that treatment-naïve MAPK pathway-mutated HNSCC tumors are the only tumors with significant elevation of CD8[+] T-cell infiltration within the tumor microenvironment (Fig 4A and B; P = 0.0030; size of red-outlined bubbles indicating significance level). Most importantly, MAPK-mutant tumors have the most remarkable "CD8[+] T-cell–inflamed status" along with most significant and concurrent increases in all three major immunoreactive signature scores, including high cytolytic process (CYT) (Rooney et al, 2015), T-effector activity (T-eff) (Bolen et al, 2017), and antitumor IFN-γ signature scores ((Ayers et al, 2017)), all evident of active, constitutive cytolytic T-cell activity native to these tumors (Fig 4A–C). Whereas for JAK/STAT- and PI3K-mutated HNSCC, only CD8[+] T-cells are increased with no obvious concurrent activation of full active CYT, T-eff, and IFN-γ signatures, suggestive of potentially weaker CD8[+] T-cell cytolytic activity in situ (Fig 4A). We also noted increases in dendritic cell (P = 0.0052) and neutrophil infiltration levels in MAPK-mutant tumors (P = 0.0235; Figs 4A and B and S9). Nonhierarchical clustering demonstrated associations between CD8[+] T-cells with immune activity signature scores and infiltrations of dendritic cells (antigen-presenting cells) and neutrophils, which were discrete from signatures for B cell, CD4[+] T-cell, and macrophage infiltrations (Fig 5A). Overall, these findings support CD8[+] T-reactive tumor microenvironments of MAPK-mutant patient tumors. All these findings are consistently noted in HPV-negative MAPK-mutated tumors as well (Figs 4A–C and S8B).

Increases of CD8+ T-cell infiltration in HNSCC tumor microenvironment alone are known to independently predict favorable patient survival (Fig 5B) (Hartman et al, 2018). Our in-house MAPK pathway-mutated HNSCC tumors also demonstrated increases in CD8[+] T-cells (accompanied by dendritic cells and neutrophil infiltrations) by immunohistochemistry, consistent with the TIMER-predicted CD8[+] T-cell–inflamed, immunoactive microenvironments borne by MAPK-mutated HNSCC tumors in TCGA (Figs 5C and S10). Figs 5C and S10 show

---

was used for Pt.25 and HSC-4 samples, 50 μg of protein lysate was used, whereas for HSC-6 (because of the relatively low endogenous p-RSK levels intrinsic to this cell line), 100 μg of protein lysate was loaded for presentation of signal clarity. Bar graphs showing the quantified changes of p-ErbB3(Y1289) levels upon GDC-0994 treatment (N ≥ 4 independent experiments). **(D)** Negative correlation between p-ErbB3(Y1289) and p-MAPK(T202/Y204) levels in MAPK-mutant HNSCC cell lines based on the published CCLE-proteomic data (Ghandi et al, 2019) and in MAPK-mutated HNSCC patient tumors (allele frequencies [AFs] >40%) based on TCPA HNSCC RPPA cohort (Li et al, 2013, 2017a). **(E)** Immunohistochemical staining for p-ErbB3(Y1289) in MAPK-mutated HNSCC patient tumors with high AFs close to 40%: T40 (*MAPK1* p.D321N with AF = 39.1%) and T43 (*HRAS* p.G12S with AF = 37.3%) versus T47 and T82 (both are MAPKWT).
Source data are available for this figure.

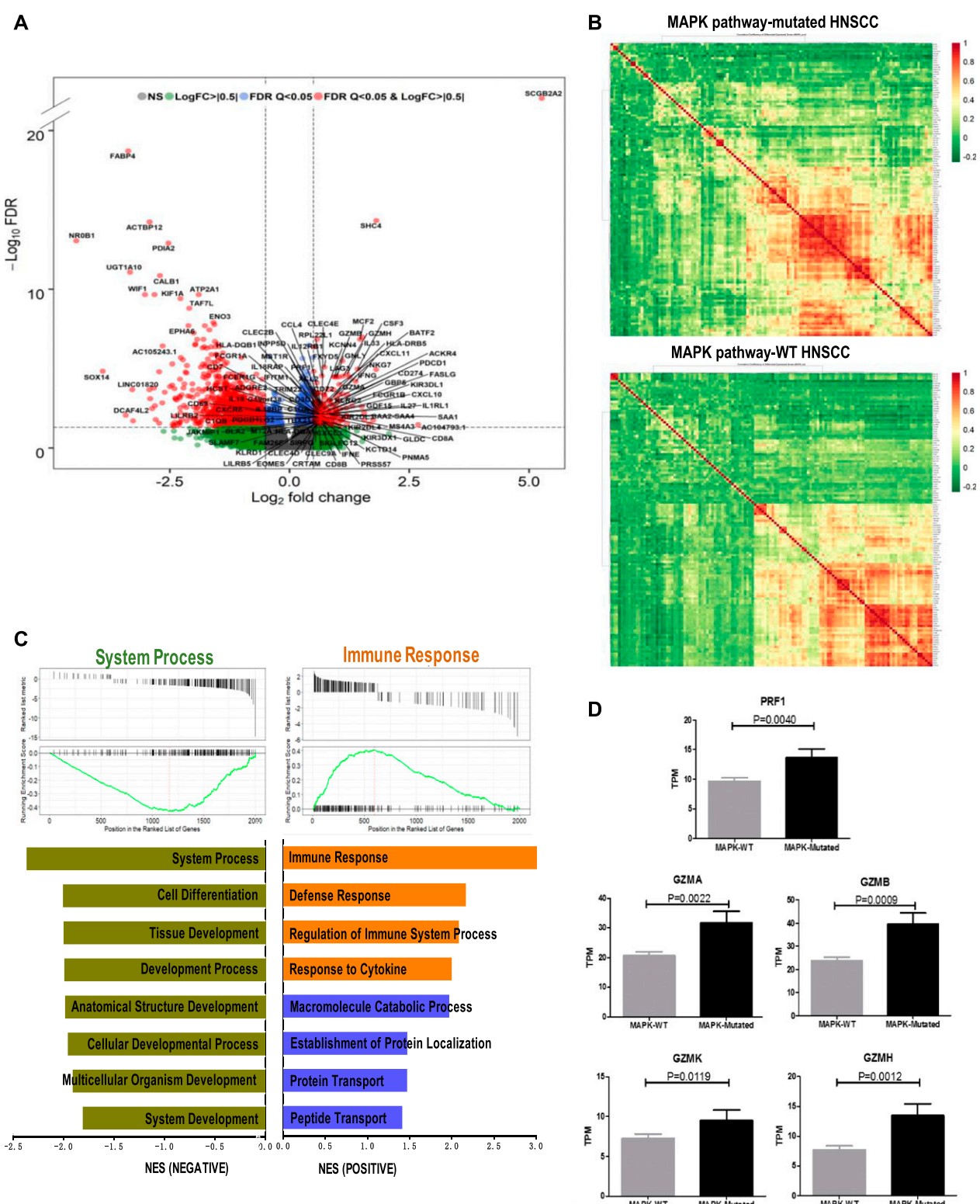

**Figure 3. Transcriptomic analyses reveal prominent immune signatures of MAPK-mutant HNSCC tumors.**
**(A)** A volcano plot showing differential RNA expressions between MAPK pathway-mutated versus WT HNSCC tumors. **(B)** Distinct gene expression patterns of MAPK pathway-mutated tumors versus MAPK-WT tumors (based on RNA-seq dataset of TCGA HNSCC cohort), with 130 protein-coding differentially expressed gene with $\log_2$ fold-change > 0.5 and false discovery rate (FDR) < 0.05 shown in the respective heat maps. **(C)** Gene Set Enrichment Analysis for MAPK pathway-mutated (versus MAPK-WT) HNSCC tumors demonstrating enrichment of immune-related gene sets in four of eight enrichment functional gene sets. **(D)** Comparison of expression levels of *PRF1*, *GZMA*, *GZMB*, *GZMH*, and *GZMK* mRNA in MAPK-mutated versus WT HNSCC tumors (based on RNA-seq dataset of TCGA HNSCC provisional cohort).

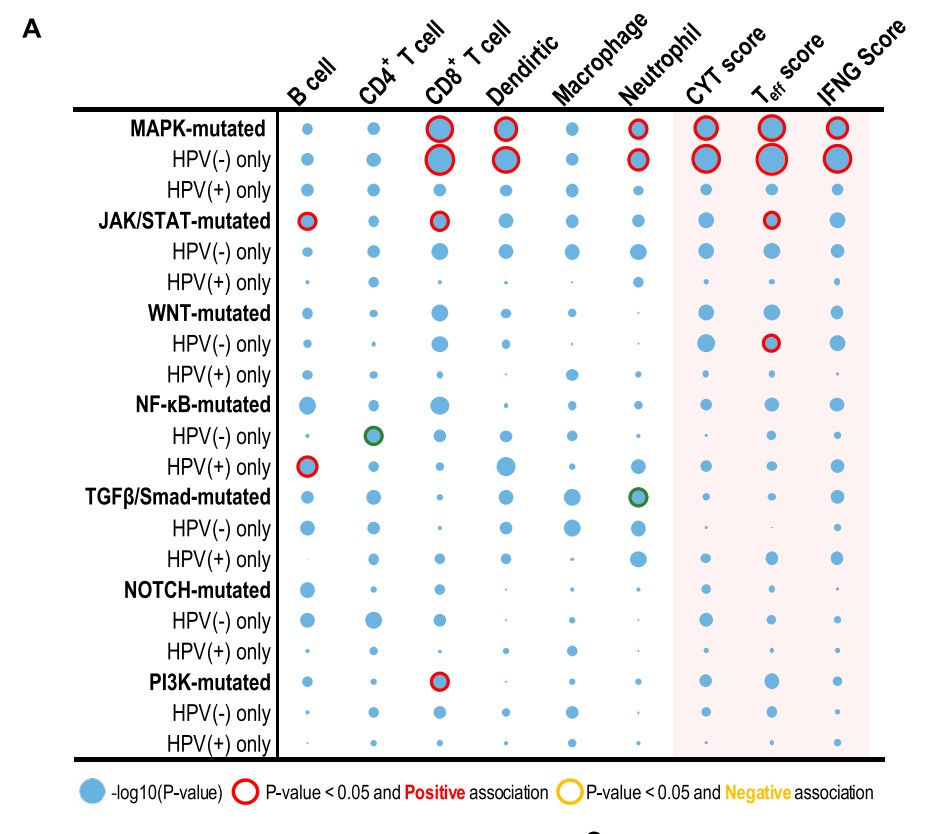

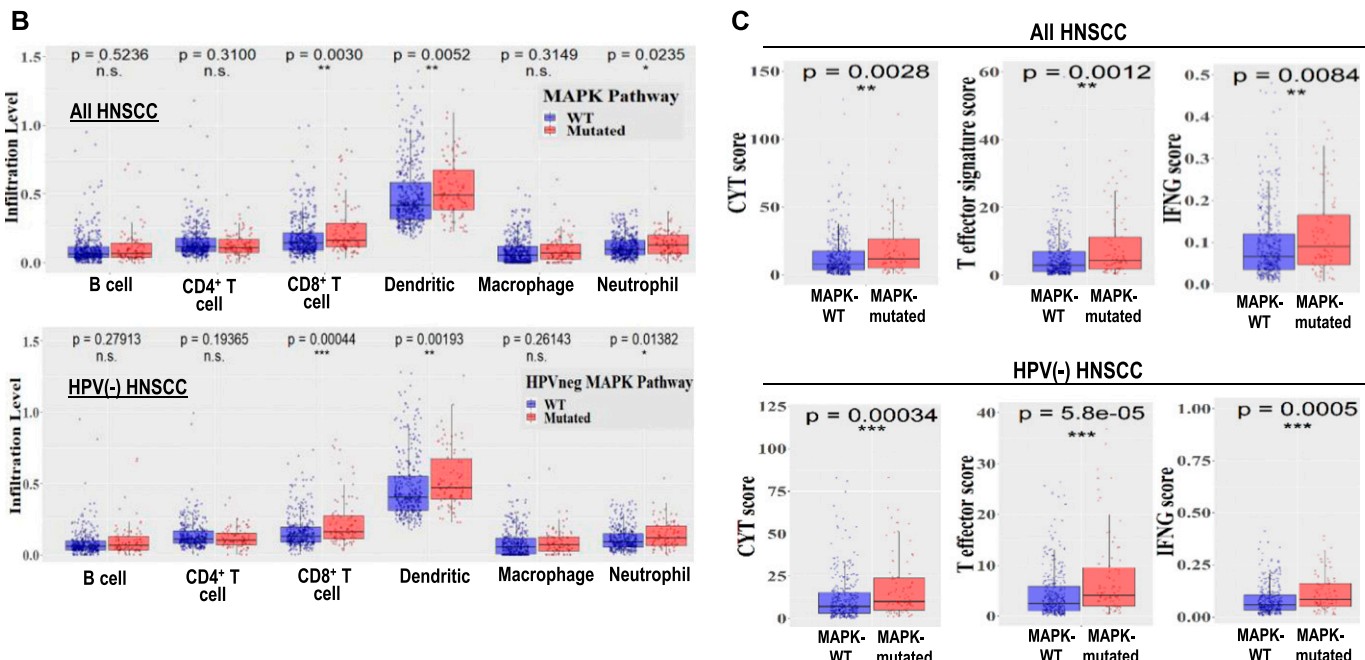

**Figure 4.   MAPK-mutant HNSCC patient tumors were CD8+ T-cell inflamed with immunoreactive cytolytic signatures.**
**(A)** A bubble plot showing the degree of statistical significance for HNSCC tumor infiltration levels of B cells, CD4+ T-cells, CD8+ T-cells, dendritic cells, macrophages, and neutrophils (by TIMER analyses (Li et al, 2016, 2017b), and the CYT score, T-effector score and IFN-γ score for HNSCC tumors bearing respective pathway mutations (versus respective WT tumors). Bubbles are highlighted in red outline when P < 0.05 with calculated positive correlations for increase in the respective TIL or immune score, and bubbles are highlighted in orange outline when P < 0.05 with calculated negative correlations indicating decrease in the respective TIL or immune score when a pathway is mutated. **(B)** Results for TIMER analysis for MAPK pathway-mutated versus WT HNSCC tumors for all HNSCC tumors (upper panel), or for human papillomavirus (HPV)-negative HNSCC only (lower panel). MAPK pathway-mutated tumor showed most significant increases in CD8+ T-cell infiltrations in all HNSCC, as well as in HPV-negative HNSCC (lower panel). P-values were shown for each immune cell type (unpaired t test). **(C)** Comparisons of CYT score, T-effector signature score, and IFN-γ functionality score between MAPK-mutated versus WT HNSCC tumors (for all HNSCC in upper panel; and for HPV-negative HNSCC only in the lower panel).

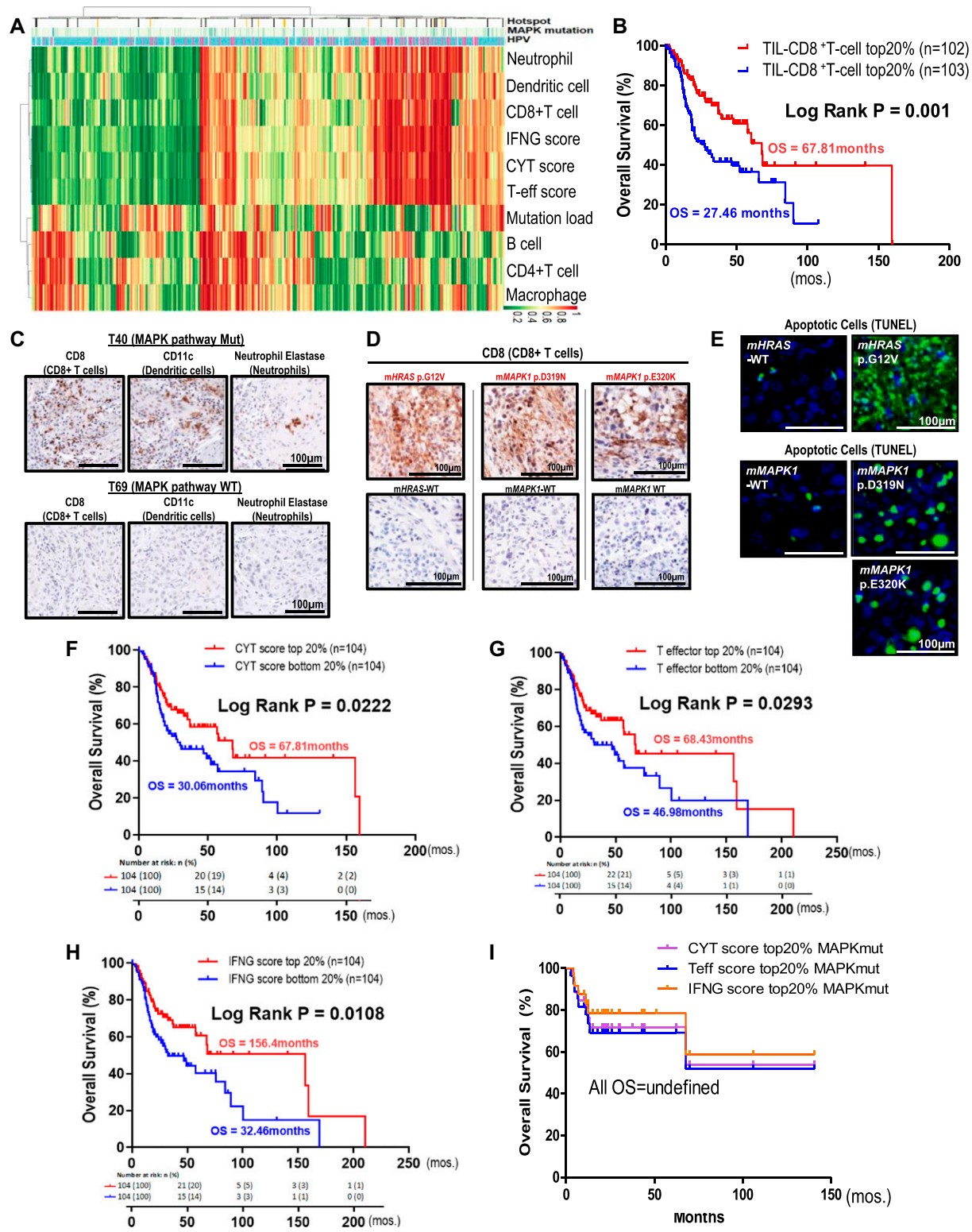

**Figure 5. CD8⁺ T-cell–inflamed and cytolytic features of MAPK-mutated head and neck squamous cell carcinoma (HNSCC) were recapitulated in immunocompetent models with massive apoptosis in situ.**

**(A)** Nonhierarchical clustering of tumor infiltrating immune cell types and immune signature scores (CYT, T-effector signature and *IFN-γ* functionality score) in TCGA HNSCC tumors with MAPK pathway mutational status and HPV status. **(B)** High infiltrating levels of CD8⁺ T-cells are associated with markedly improved HNSCC patient survival compared with patients with low CD8⁺ T-cell infiltration levels (top and bottom 20% cutoffs). **(C)** Immunohistochemical staining of MAPK pathway-mutated HNSCC tumors (T40 with *MAPK1* p.D321N hotspot mutation) showing increased expressions of the CD8 marker (for CD8⁺ T infiltration). These tumors also expressed higher

immunohistochemical validation of these heavy immune infiltrations in MAPK-mutant tumors (T40 with *MAPK1* p.D321N, T06 with *HRAS* p.G13D, T25 with *HRAS* p.G12S, and *MAPK1* p.R135K), whereas such infiltrates were not detected in WT tumors (T69, T75, and T39).

## Immunocompetent MAPK-mutant HNSCC models display active cell death coupled with massive CD8⁺ T-cell recruitment in situ

As recently recognized, increases in intratumoral CD8⁺ T-cell infiltration in HNSCC dictate patient outcomes (Fig 5B) (Hartman et al, 2018), we determined the ability of MAPK mutations to induce CD8⁺ T-cell infiltration in immunocompetent HNSCC models. Mouse counterparts of three most common human MAPK pathway mutations, and respective WT controls, were expressed into a mouse HNSCC cell line (SCC VII), and their abilities to induce intratumoral CD8⁺ T-cell infiltration in vivo were compared. As shown in Fig 5D, all three mutants with HNSCC relevance, murine *HRAS* p.G12V (*mHRAS* p.G12V), *mMAPK1* p.D319N (equivalent to human p.D321N), and *mMAPK1* p.E320K mutations (equivalent to human p.E322K), were all potent recruiters for CD8⁺ T-cell infiltration in situ. Furthermore, consistent with our findings that MAPK-mutated human HNSCC tumors have inherently high active cytolytic signatures (Figs 3D and 4A–C), we also observed remarkable increases in mutant-specific apoptosis in vivo (a two to eight times increase in TUNEL positivity in mutant versus WT tumors; Fig 5E). Our data first established these major MAPK mutations as direct and potent inducers of CD8⁺ T concentration in HNSCC models of isogenic background. Our findings are consistent with an inherently active high cytolytic tumor microenvironment identified in TCGA human MAPK-mutant HNSCC tumors.

## Immunoactive and ErbB3-inhibitory activities of MAPK-mutated HNSCC are independent

In addition to high CD8⁺ T-cell infiltration, we also identified that very high immune scores, that is, high IFN-γ score, high CYT score, and high T-eff signature score (top 20% versus bottom 20% arbitrary cutoffs) were all survival-favoring features among HNSCC patients (Fig 5F–H; versus patients with tumors of respective low scores). These patients also have reduced chances of death (odds ratios = 0.4949, 0.5446, and 0.5889, respectively, *P* < 0.05, Fisher's exact test). Importantly, MAPK pathway-mutated HNSCC patients bearing high IFN-γ scores, high CYT scores, and high T-eff scores have long-term survivals (median not reached; Fig 5I). Patients with high immune scores (IFN-γ, CYT, and T-eff), defined by extreme cutoffs of quintile,

quartile, and tertile (top 20%, 25%, and 33% versus bottom 20%, 25%, and 33%, respectively) were almost all significantly associated with better patient survival (except for a demonstrated trend for quartile cutoff with CYT score, *P* = 0.083). At the most relaxed median cutoff, both the IFN-γ and T-eff scores could still separate patient survival (*P* < 0.05), but not CYT score (*P* = 0.2) (Figs S11–S13).

Notably, among MAPK-mutated HNSCC patients, we found that those with high IFN-γ, CYT, or T-eff scores (both at 20% and 50% cutoffs) do not significantly overlap with patients of low p-ErbB3 (P = n.s., Fig 6A), suggesting two independent mechanisms operative to contribute the improved outcomes of MAPK pathway-mutant HNSCC. The overall mechanistic findings are summarized in Fig 6B.

## MAPK mutations predict immunotherapy outcomes in pan-cancers, oral cancer, and metastatic HNSCC

Samstein et al (2019) recently published the largest clinical dataset for immune checkpoint inhibitors (ICIs: PD1/PD-L1 and CTLA4 inhibitors) with treatment outcomes in 1,662 advanced or metastatic cancer patients across 11 cancer types. The study concluded that high TMB (high defined as top 20% patients within each cancer type) could predict ICI responders in most cancer types (Samstein et al, 2019). Unexpectedly, MAPK pathway mutations alone predict almost two times longer survival in pan-cancer patients upon ICI treatment as compared with MAPK-WT patients (31 versus 16 cutoffs mo, *P* < 0.0001; Fig 7A). Note that in this target sequencing ICI study, only 10 MAPK members were sequenced. Importantly, >81% of pan-cancer patients were TMB-low (Fig 7B). Strikingly, we found that among TMB-low patients, MAPK pathway mutations were also significantly prognostic for favorable survivals with an OS of 25 versus 14 mo in WT patients (*P* = 0.0106, Fig 7C). However, MAPK mutations do not predict survival among TMB-high patients (P = 0.5091). Importantly, our novel finding that MAPK pathway mutations could predict favorable outcomes in patients treated with ICI was further cross-validated in an independent whole-exome sequenced multi-cancer cohort from Miao et al (2018) (Fig 7D).

CD8⁺ T-cell infiltration in the tumor microenvironment are believed to be critically involved in the clinical activity of ICIs (Melero et al, 2014). As per our finding that MAPK-mutated HNSCC tumors are endowed with inherently high CD8⁺ T-cell–inflamed immunoactive tumor microenvironment with high endogenous cytolytic activity, we analyzed HNSCC patient outcomes from the Samstein-ICI HNSCC dataset. For this HNSCC dataset, oral cancer (N = 47) and oropharyngeal cancer (N = 32) were two major cancers with sufficient case numbers for survival analysis. Strikingly, among advanced oral

---

levels of CD11c marker (for dendritic cell infiltration) and neutrophil elastase marker (for neutrophil infiltration) when compared with MAPK-WT tumors (T69). Scale bars are shown. **(D)** Increased expressions of CD8 marker (for CD8⁺ T infiltration) were detected in *mHRAS* p.G12V mutant, *mMAPK1* p.D319N mutant (corresponding to *MAPK1* p.D321N mutation in human *MAPK1*), and *mMAPK1* p.E320K mutant (corresponding to *MAPK1* p.E322K mutation in human *MAPK1*), and in their respective mouse WT xenografts by immunohistochemistry on day 6 (for m*MAPK1* WT versus m*MAPK1* p.E320K pair) and on day 11 (for m*HRAS* WT versus m*HRAS* p.G12V & m*MAPK1* WT versus m*MAPK1* p.D319N pairs) after tumor cell inoculation. **(E)** Dramatic increases of apoptotic cells were also observed in MAPK pathway-mutated tumors labeled with TUNEL and corresponding DAPI staining on day 6 after tumor cell inoculation. **(F, G, H)** Kaplan–Meier overall survival (OS) curves for HNSCC patients with (F) higher IFN-γ functionality score, (G) higher T effector signature score, and (H) higher CYT score and versus patients with respective lower *IFN-γ* functionality, T-effector signature and CYT score in TCGA HNSCC (N = 522) RNA-seq cohort (top 20% and bottom 20% cutoffs). **(I)** Kaplan–Meier OS curves for MAPK pathway mutated patients with high *IFN-γ* functionality score and high T-effector signature score and high CYT score (all top 20% cutoffs), demonstrating long OS (median OS not reached).

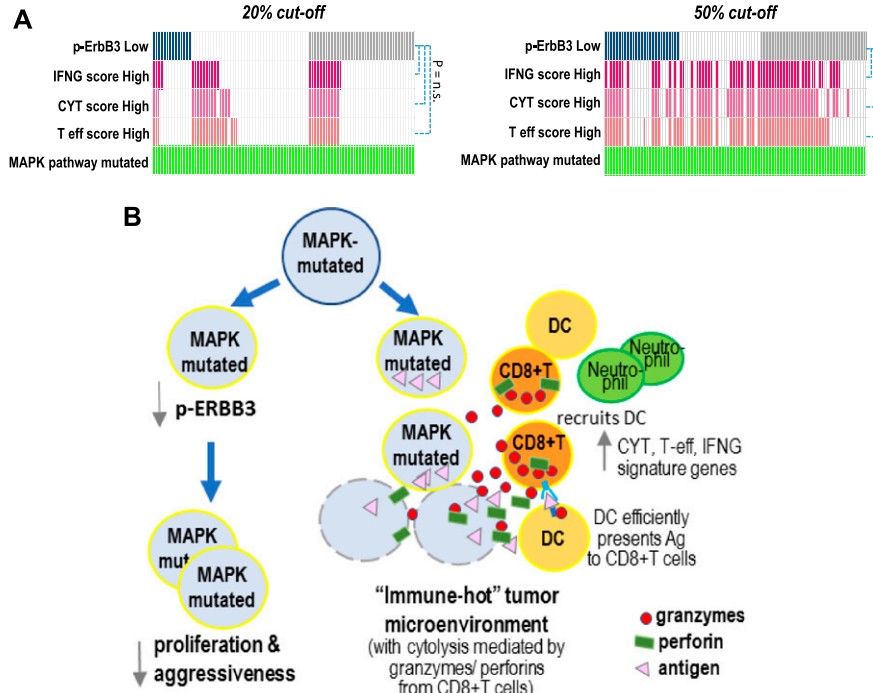

**Figure 6. ErbB3 inhibition and CD8⁺ T-cell immunoactivation as plausible mechanisms contributive to improved survival of MAPK pathway–mutated HNSCC patients.**
**(A)** Oncoprints (20% and 50% cutoffs respectively) showing that MAPK pathway-mutated patients with low p-ErbB3 protein expression, and high IFN-γ functionality, T-effector signature and CYT score were not significantly overlapping (P = n.s.). The upper panel shows the oncoprint with deep blue color ( ) denoting those with bottom 20% of pErbB3 level (i.e., p-ErbB3 down-regulation), and levels above that as noncolored white bars ( ), whereas individuals without available RPPA data on p-ErbB3 are denoted by grey bars ( ). Similarly, those with top 20% immune scores, IFN-γ score, CYT score and T-effector are indicated by deep pink ( ), light pink ( ) and orange ( ), respectively, while non-colored white bars ( ) denote patients with immune scores lower than the top 20%. In the lower panel, the same color coding is adopted, but the colored bars refer to patients with a median (i.e., 50% cutoffs), that is, lower 50% for p-ErbB3 level and top 50% for IFN-γ, CYT, and T-effector scores. **(B)** A schematic summarizing two plausible mechanisms for markedly improved clinical outcomes in HNSCC tumors with MAPK aberrations.

cancer patients treated with PD1/PD-L1 inhibitors, MAPK pathway mutations appeared to predict a 3.3 times longer median OS versus WT (33 versus 10 mo, P = 0.0466; Fig 7E; all tumors with >10% tumor purity), supportive of a CD8⁺ T-cell–inflamed phenotype for improved ICI outcome. Whereas TMB-high status only demonstrated a trend for better outcome (Fig S14A), consistent with recent clinical findings that in HNSCC, TMB status may not accurately predict patient outcome as compared with PD-L1 status (Cohen et al, 2019). Importantly, we consistently found that MAPK-mutant oral cancers are significantly associated with immune class defined by Chen et al (2019), with an odds ratio of 0.531 in oral cavity dataset (N = 309, P = 0.019 Fisher's exact test) (Fig 7F). This was not observed in the small oropharyngeal cancer dataset in which no HPV status was recorded (Fig S15).

Notably, as high as 49% of HNSCC tumors (53/110) sequenced by Samstein et al (2019) were distant metastases (lung, liver, heart, brain, and bone). Strikingly, among this very worst prognostic subgroup with distant metastases, we found that MAPK mutations in metastatic lesions also predicted an approximately four times longer OS versus WT patients upon PD1/PD-L1 inhibitor treatments (median OS of 33 versus 8 mo; Fig 7G). Contrarily, TMB-high status did not predict HNSCC survival (Fig S14B, P = n.s.). Furthermore, in both oral cancer and distant metastatic HNSCC, MAPK-mutant and TMB-high subsets of patients were not overlapping (P = n.s.; Fig 7H and I), demonstrating a TMB-independent predictive power of MAPK pathway mutations in HNSCC settings.

Overall, MAPK pathway mutations may represent novel and important HNSCC biomarkers for prognosis in HNSCC. Based on our transcriptome findings, validated in immunocompetent models, MAPK mutations likely identify highly CD8⁺ T-cell–inflamed/cytolytically active HNSCC patients who may potentially benefit from ICIs. Our study uncovers novel clinical,

biological, and immunological uniqueness of MAPK-mutant HNSCC and may indicate wide clinical utilities of MAPK pathway mutations in this cancer.

# Discussion

We reported for the first time that MAPK pathway mutations have significant survival-favoring clinical, signaling, and immunological impacts in HNSCC. Among seven major cancer-signaling pathways examined, MAPK (ERK) pathway mutations, predominated by many activating mutations, were prognostic for remarkably long patient survivals. This is contradictory to the known role of MAPK-mitogenic signaling in HNSCC tumorigenesis and progression (Gkouveris et al, 2014; Lakshmanachetty et al, 2019). This finding was cross-validated in an independent MSK-IMPACT HNSCC cohort. Furthermore, the prognostic power of MAPK pathway mutations is independent of HPV. Similar superior survival was also observed in MAPK-mutated en-dometrial cancer patients (TCGA). The finding that MAPK pathway mutations span the diversity of HNSCC subsites (including oral, oropharyngeal, laryngeal, and pharyngeal) to identify patients with good outcomes is superior to that of HPV-positivity (largely restricted to oropharyngeal subsite), and in particular, these mutations predict extraordinary patient outcomes even among TP53-mutated HNSCC patients (median OS >14 yr). Thus, MAPK mutations may represent novel prognostic or even de-intensification biomarkers for HNSCC with broad applicability in terms of HNSCC subsites.

Importantly, we also first identified that HNSCC patient tumors with high allele frequencies of MAPK-activating mutations, such as

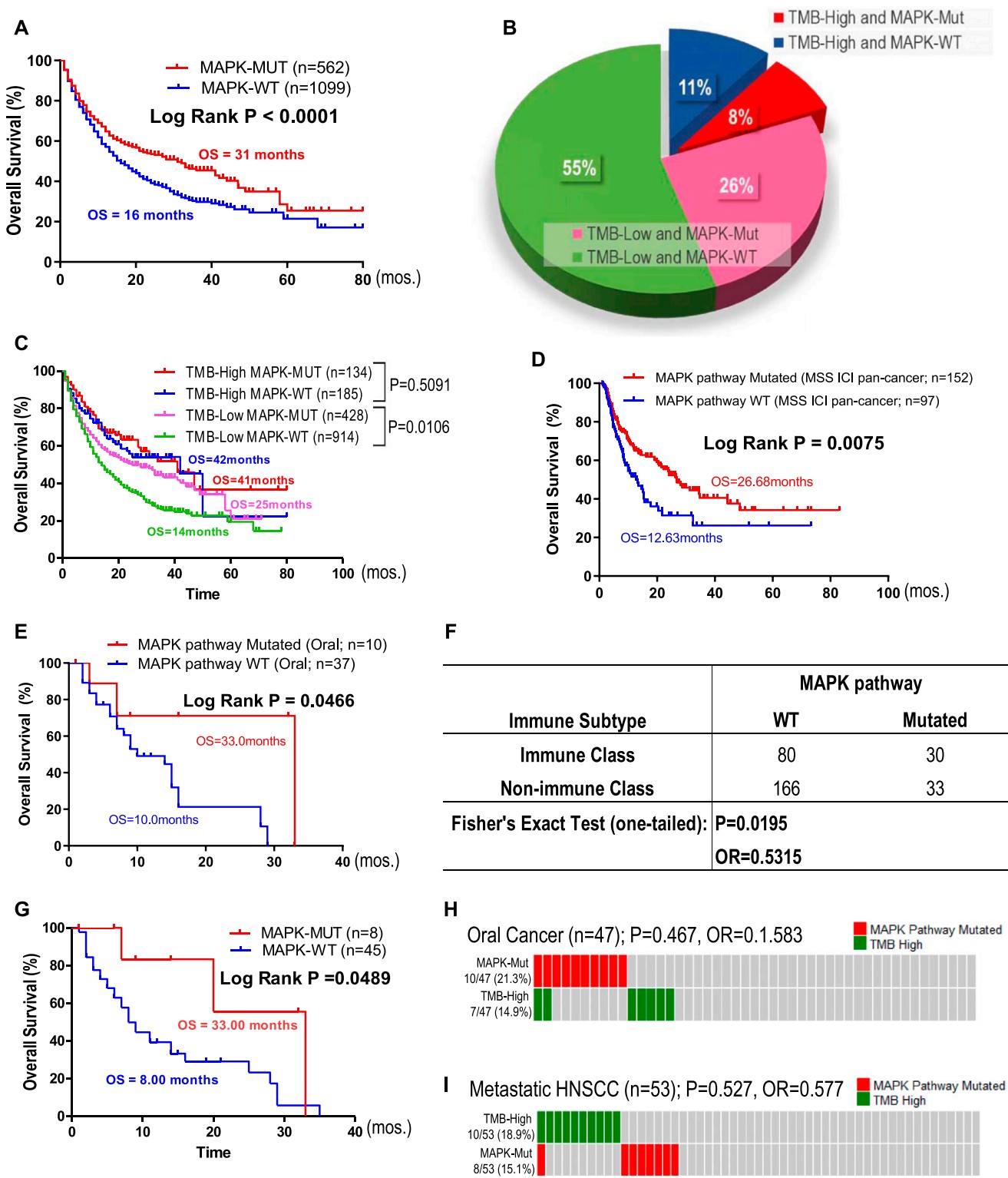

**Figure 7. MAPK pathway mutations may predict patient outcomes with immune checkpoint inhibitors independent of tumor mutational burden (TMB).**
**(A)** MAPK pathway mutations (10 genes) are associated with good clinical outcome from an independent cohort of Samstein et al (2019) in advanced or metastatic pan-cancer patients (N = 1,662) treated with PD1/PD-L1 or CTLA4 inhibitors. **(B)** A pie chart showing 34% patients with somatic MAPK pathway mutations in both TMB-high and TMB-low groups of patients in the pan-cancer dataset. High-TMB was previously defined as top 20% cutoff (i.e., TMB value ≥10.3 for head and neck squamous cell carcinoma [HNSCC]), whereas low-TMB represented the remaining 80% of patients (i.e., TMB value < 10.3 for HNSCC) per original publication by Samstein et al (2019). **(C)** Overall survival (OS) curves of four subgroups of patients: TMB-high with MAPK mutations, TMB-high with MAPK-WT, TMB-low with MAPK mutations, and TMB-low with MAPK-WT in

HRAS and MAPK1 hotspot mutations and potentially others, have remarkable tumoral ErbB3-suppression (first identified in TCGA data and validated in HNSCC tumors). Mechanistically, our findings identified a previously undescribed mechanism of p-ErbB3 regulation by MAPK pathway-mutants. Such a negative regulation of p-ErbB3 by ERK activity is uniquely found in MAPK-mutant, but not in MAPK-WT HNSCC. Most importantly, MAPK-mutant HNSCC tumors are the only tumors having significant "CD8⁺ T-cell–inflamed" and inherently immunoactive tumor microenvironment (versus six other pathway-mutant tumors), with constitutive cytolysis. The ability of MAPK mutations to drive a CD8⁺ T-cell–inflamed status in vivo with marked apoptosis is proven in immunocompetent HNSCC models. As low tumoral phospho-ErbB3 levels and elevated CD8⁺ T-cell infiltrations are recently established events indicative of good patient survivals in HNSCC (Takikita et al, 2011; de Ruiter et al, 2017), our study first defined somatic MAPK pathway mutations as novel genomic events governing two outcome-favoring features in HNSCC. We further showed that the ErbB3-suppressive and CD8⁺ T-cell–inflamed tumor microenvironments of MAPK pathway mutant HNSCC tumors are likely two independent molecular characteristics of MAPK-mutated HNSCC patients with remarkably improved outcomes.

Our findings not only enrich our understanding of the immune uniqueness of MAPK-mutated HNSCC tumors in patients but also highlights the potential clinical utility of MAPK pathway mutations in identifying HNSCC patients with CD8⁺ T-cell–inflamed tumors, independent of TMB, for likely beneficial PD1/PD-L1 inhibitor treatments. This positive prognosticity of MAPK pathway mutations may potentially be beneficial to pan-cancer, as shown in two independent immunotherapy cohorts. In conclusion, our study uncovers novel clinical, biological, and immunological understanding of MAPK pathway mutations in HNSCC, which may have important clinical impacts on HNSCC management as prognostic biomarkers and as predictive biomarkers for potential immunotherapy benefits.

# Materials and Methods

## Pathway component definitions and databases used

The MAPK pathway was defined as *H/K/N-RAS, A/B-RAF, RAF1, MAP2K1/2* (*MEK1/2*), *MAPK1/3*(*ERK2/1*), *RPS6KA1, SHC1/2/3/4, GRB2, and* Erk1/2-*specific DUSP3/5/6/7/9*. The PI3K pathway was defined as *AKT1/2/3, PIK3CA/B/D/G/2A/2B/2G, PIK3AP1, PIK3IP1, PDK1, MTOR, TSC1, TSC2, PTEN, RICTOR, RPTOR, RHEB,* and *PIK3R1/2/3/4/5/6*. The NF-κB pathway was defined as *TAB1/2/3, MAP3K7/14, CHUK,*

*IKBKB, IKBKG, NFKBIA, NFKBIE, REL, RELA/B, NFKB1/2, LTBR, TNF, TNFAIP3, TNFSF11/13B, TNFRSF1A/8/11A/13C, BTRC, CYLD, NLRC5, TRADD, CD40, CD40LG, LTA, TRAF2/3/5/6, IL1B,* and *IL1R1*. The JAK/STAT pathway was defined as *JAK1/2/3, STAT1/2/3/4/5A/5B/6, PTPN11, IL6, IL6R, IL6ST,* and *SOCS3*. The Notch pathway was defined as *DLL1/3/4, JAG1/2, NOTCH1/2/3/4, NUMB, DTX1/3L, NEDD4, MAML1, RBPJ, POFUT1, HES1/5,* and *HEY1/2/L*. The WNT pathway was defined as *WNT1/3A/5A/5B/7A, CTNNB1, HNF1A, FZD1/2/3/7/8/9/10, AXIN1, LEF1, LOXL2, DVL2/3, NKD1/2, TAB1/2, GSK3B, CSNK1A1, NLK,* and *LRP5/6*. The TGF-β/Smad pathway was defined as *SMAD1/2/3/4/5/6/7/9, TGFB1/2/3, TGFBR1/2, INHBA/B/C/E, NODAL, ACVR1/1B/1C/2A/2B, BMP2/4/7, BMPR1A/B/2, AMHR2, LTBP1, BAMBI, ZFYVE9, SMURF1/2,* and *LIMK1*. All TCGA whole-exome sequencing data and clinical data of TCGA are downloaded from the www.cbioportal.org (Cerami et al, 2012; Gao et al, 2013) on 28 November, 2018. Whole-exome sequencing, RNA-seq, and clinical survival data are available for 510, 522, and 527 (397 for disease-free) HNSCC cases, respectively. Protein quantitative expression levels were downloaded from TCPA level 4 data on 21 June, 2018 (Li et al, 2013, 2017a). Kaplan–Meier survival curves are plotted with GraphPad Prism 5 (USA), with calculated log-rank test P-values. The favorable/unfavorable OS indicated in Fig 1B and E are defined by log-rank P-value (P < 0.05) generated from Kaplan–Meier curves as in Fig 1C and D. In Fig 1B, patients are stratified by the specified status, for example, MAPK pathway mut versus MAPK pathway WT, PI3K pathway mut versus PI3K pathway WT, HPV(+) versus HPV(−), etc. Then, the OS advantages are being analyzed with Kaplan–Meier survival analyses. Only those patient groups with log-rank P-value < 0.05 (significant) would we consider them having favorable/unfavorable OS. CCLE-proteomic database was downloaded from DepMap portal (http://www.depmap.org) as published by the CCLE study (Ghandi et al, 2019), and immunotherapy clinical database and targeted sequencing database were downloaded from the studies by Samstein et al (2019) and Miao et al (2018).

## Cell cultures and drug treatment

FaDu cells and HSC-4 cells were purchased from ATCC and JCRB, respectively. Cell lines around passages 10–25 were used (within 6 mo of purchase, which were mycoplasma free when purchased). The Platinum-A (PLAT-A) retrovirus packaging cell line was purchased from Cell Biolabs. The HSC-6 cell line was a generous gift from Dr J Inazawa (Tokyo Medical and Dental University, Japan), and SCC VII mouse HNSCC cell line was kindly provided by Dr Sven Branduau (University Hospital Essen, Germany). Pt-25 primary cultures were prepared from a female recurrent HNSCC patient. For GDC-0994 treatment, the cells were plated at ~30% confluency for

the Samstein study. **(D)** Kaplan–Meier OS curves for MAPK pathway-mutated HNSCC patients versus MAPK pathway WT patients (MSS ICI pan-cancer cohort; N = 249). **(E)** Kaplan–Meier OS curves for MAPK pathway-mutated HNSCC patients versus MAPK pathway WT patients (study by Samstein et al (2019); HNSCC Ooal subsite cohort; N = 47). **(F)** Table of Fisher's exact test showing association of HNSCC subtypes with the immune class as defined by Chen et al (2019). **(G)** In HNSCC patients with distant metastases (lung, liver, heart, brain, and bone), MAPK pathway mutations are associated with better OS upon PD1/PD-L1 inhibitor treatment (P = 0.0489; based on databases from Samstein et al (2019)). **(H, I)** The corresponding oncoprints showing no significant overlap between (H) patients with MAPK pathway mutations and high tumor mutational burden in this HNSCC-oral cancer cohort (TMB score 20% cutoff within HNSCC histology [N = 139] in the study by Samstein et al (2019)) and (I) no significant overlap for patients with MAPK pathway mutations and high tumor mutational burden in this HNSCC distant metastasis cohort (TMB score 20% cutoff within HNSCC histology).

overnight and then subjected to either vehicle or 0.5 $\mu$M of GDC-0994 for 30 min. The cells were then washed with 1× PBS, and protein lysates were prepared for Western blot analyses.

## Retroviral introduction of pathway genes/mutants into HNSCC models

The retroviral Plat-A amphotropic expression system (Cell Biolabs, Inc.) was used for ectopic expression of genes and mutants into HNSCC cells. In brief, the desired genes/mutants cloned into the pMXs-puro retroviral expression vector backbone were transfected into PLAT-A cells using Lipofectamine 3000 (Thermo Fisher Scientific) for the generation of retroviruses. Retroviruses were filtered through a 0.45-$\mu$m mixed cellulose ester membrane filter to remove cell debris and subsequently used for infection of FaDu cells or SCC VII cells for 48–72 h at 37°C, 5% $CO_2$. Retroviruses were removed from the infection medium at postinfection, and cells were then cultured in their respective culture media. Gene expression were then validated by Western blotting. For mouse HNSCC tumor cell inoculation (SCC VII, originally derived from C3He mouse background [Suit et al, 1985]), 2–3 million cells expressing each gene or the respective mutant were subcutaneously injected into females of the C3H/He sub-strain (C3HeB/FeJLe-a) mouse strain for tumor establishment with a 26G Hamilton syringe. Tumors were harvested at the designated days for tumor collection. All animal experiments were approved by the University Animal Experimentation Ethics Committee of the Chinese University of Hong Kong.

## Western blotting

Cells were washed with cold 1× PBS and lysed with the NP40 lysis buffer (1% Nonidet-P40, 150 mM NaCl, 1 mM EDTA, 10 mM sodium phosphate buffer, and protease phosphatase inhibitor). Cell lysates were centrifuged, and the supernatant was quantified with Protein Assay Dye Reagent (Bio-Rad). 50 μg of protein lysates were mixed with a 4× protein loading dye and then separated by SDS–PAGE. Separated proteins were transferred to nitrocellulose membrane, which was then blocked with 5% nonfat dry milk (in TBST; 150 mM NaCl, 50 mM Tris, and 0.1% Tween 20, pH 7.4) and probed with primary antibody at 4°C overnight. Primary antibodies include AKT (#9272), pi-AKT (#9271), ARAF(#4432), pi-ARAF(S299) (#4431), BRAF (#2696), pi-BRAF(S445) (#2696), pi-ErbB3 (#2842), ErbB3 (#12708), MAPK (#9102), pi-MAPK (#9101), pi-MEK1/2 (#9154), and RSK1 (#8408), all from Cell Signaling Technology, USA. Anti-$\beta$-actin (sc-69879) antibody was from Santa Cruz. Anti-MEK1/2 (YT2714) and anti-N/H/K-RAS (YT2960) antibodies were from ImmunoWay. The probed membrane was then washed three times with 1× TBST, followed by a 1–2-h incubation with the respective secondary antibody (Goat anti-Mouse [ab97230; Abcam] and HRP-Goat anti-Rabbit [65-6120; Invitrogen]), followed by 3× washings with 1× TBST. ECL detection solution was then applied onto the membrane for the development of chemiluminescence, which was captured by autoradiography.

## TIMER analysis for immune infiltrates

We adopted the methodology of TIMER analysis for immune infiltration level estimation (Li et al, 2016, 2017b). The infiltration level of six immune cell types (B cell, CD8[+] T-cell, CD4[+] T-cell, macrophage, neutrophil, and dendritic cell) in 512 TCGA-HNSCC head and neck squamous carcinoma tumor samples are extracted from the TIMER Web site (https://cistrome.shinyapps.io/timer/_w_20aca96c/immuneEstimation.txt) and grouped according to the mutational information downloaded from cBioPortal (https://www.cbioportal.org/study/summary?id=hnsc_tcga). The Wilcoxon rank sum test was performed to calculate the statistical significance between two groups in the TIMER plots.

## RNA-seq and GSEA

TCGA HNSCC RNA-seq data were downloaded from the NCI Genomic Data Commons portal (https://portal.gdc.cancer.gov/repository?facetTab=files&filters=%7B%22op%22%3A%22and%22%2C%22content%22%3A%5B%7B%22op%22%3A%22in%22%2C%22content%22%3A%7B%22field%22%3A%22cases.project.project_id%22%2C%22value%22%3A%5B%22TCGA-HNSC%22%5D%7D%7D%2C%7B%22op%22%3A%22in%22%2C%22content%22%3A%7B%22field%22%3A%22files.data_category%22%2C%22value%22%3A%5B%22Transcriptome%20Profiling%22%5D%7D%7D%5D%7D&searchTableTab=files) (Grossman et al, 2016) using GDCRNATools (R package). Differential gene expression analysis between MAPK-mutated and MAPK-WT samples was performed using the method of DESeq2. Pearson correlation coefficients among 130 DEGs were calculated for the plotting of the correlation heatmap with pheatmap (R package) using the wardD clustering method. GSEA was run with the clusterProfiler (R package).

## Cytolytic score, T-effector signature score, and antitumor IFN-γ score calculations

Cytolytic score (CYT) was calculated by the geometric mean of the TPM of *GZMA* and *PRF1* as previous described by Rooney et al (2015) (offset 0.01) (Rooney et al, 2015). Similarly, based on the T-effector signature gene list, T-effector (T-eff) score was calculated as the geometric mean of the TPM of *GZMA*, *GZMB*, *PRF1*, *IFN-γ*, *EOMES*, and *CD8A* (offset 0.01) ((Bolen et al, 2017)). Antitumor IFN-γ score was the weighted arithmetic mean of the TPM of 18 IFN-γ expanded immune gene signatures as previously described by (Ayers et al, 2017). These include *CD3D*, *IDO1*, *CIITA*, *CD3E*, *CCL5*, *GZMK*, *CD2*, *HLA-DRA*, *CXCL13*, *IL2RG*, *NKG7*, *HLA-E*, *CXCR6*, *LAG3*, *TAGAP*, *CXCL10*, *STAT1*, and *GZMB*.

## Tumor samples and targeted sequencing

Tumor tissues and blood samples were collected from patients under written informed consents according to clinical research approvals by the Institutional Review Board of the University of Hong Kong/Hospital Authority Hong Kong East Cluster Research Ethics Committee (for Queen Mary Hospital), the Joint Chinese University of Hong Kong–New Territories East Cluster Clinical Research Ethics Committee (for Prince of Wales Hospital), Hong Kong SAR, and the Research Ethics Committee, Kowloon West Cluster (for Yan Chai Hospital), Hong Kong SAR. Genomic DNA from samples were extracted with the DNeasy Blood & Tissue Kit (QIAGEN), followed by quantification and targeted sequencing by next-generation sequencing using the IonS5 platform (Thermo Fisher Scientific). All samples were sequenced with a mean depth of >500× using a

Table showing the sequenced locations and the aminoacids convered of the partially sequenced genes in our custom-designed MAPK pathway gene panel.

| Gene | Chromosome | Locus position | Amino acid position |
|---|---|---|---|
| *ARAF* | ChrX | 47426003-47426252 | 186-233 |
| *BRAF* | Chr7 | 140434532-140434640, 140439596-140439815, 140449044-140449156, 140449186-140449273, 140453021-140453215, 140453948-140454071, 140476652-140476772, 140476828-140476946, 140477788-140477937, 140481362-140481448 | 454-526, 545-631, 642-667, 679-722 |
| *HRAS* | Chr11 | 533782-533882, 534220-534306 | 6-34, 58-91 |
| *KRAS* | Chr12 | 25368427-25368555, 25378527-25378605, 25378635-25378758, 25380260-25380368, 25398189-25398310 | 4-66, 98-121, 132-173 |
| *MAPK1* | Chr22 | 22123469-22123664, 22127153-22127280, 22142514-22142695, 22142879-22143107, 22153230-22153505, 22160105-22160333, 22161922-22162142, 22221511-22221644 | 29-362 |
| *MAP2K1* | Chr15 | 6667947-66679845, 66727308-66727523, 66727541-66727651, 66728983-66729286, 66735605-66735722, 66736897-66737080, 66774071-66774222, 66777239-66777548, 66779529-66779647, 66781535-66781633, 66782015-66782132, 66782829-66782961 | 21-356 |
| *MAP2K2* | Chr19 | 4090254-4090351, 4090679-4090789, 4094494-4094602, 4095288-4095521, 4097192-4097295, 4097316-4097449, 4099378-4099495, 4101127-4101237, 4101257-4101387, 4102338-4102451, 4110387-4110512, 4110586-4110696, 4117321-4117447, 4117464-4117566 | 52-85, 92-124, 149-183, 191-198, 236-246, 308-315, 323-350, 365-373, 483-493 |

custom-designed gene panel on major MAPK pathway genes. The custom gene panel consisted of amplicons that covered all exons of *MAP2K1* and *MAPK1*. It also captured selection regions of *ARAF*, *BRAF*, *HRAS*, *KRAS*, and *MAP2K2* to cover all well-known hotspot sites. The sequenced locations for these partially sequenced genes are listed in the following table, and variants were called by the Ion Reporter Software (Thermo Fisher Scientific).

### Immunohistochemistry and TUNEL assay

Patient tumors were freshly fixed with 10% formalin and dehydrated in a serial manner in ethanol. The formalin fixed and paraffin embedded tumor samples were then sectioned and dewaxed and rehydrated in xylene, 100% ethanol, 70% ethanol, and running tap water. Antigen retrieval was performed at 95°C for 20 min in citrate buffer (10 $\mu$M citrate acid and 0.05% Tween 20, pH 6.0). The VECTASTAIN Elite ABC Universal PLUS Kit Peroxidase (Horse Anti-Mouse/Rabbit IgG) (Cat. no. PK-8200) was used for immunohistochemical staining. Endogenous peroxidase activity was quenched by BLOXALL Blocking Solution, followed by blocking in 2.5% Normal Horse Serum for 20 min at room temperature. CD8 mouse anti-human antibody (Cat. no. Ab17147, 1:500; Abcam), CD11c rabbit antihuman antibody (Cat. no. Ab52632, 1:500; Abcam), and neutrophil elastase rabbit antihuman antibody (Cat. no. Ab68672, 1:500; Abcam) and phospho-HER3/ErbB3(Tyr1289) (21D3) rabbit antihuman antibody (#4791, 1:100; Cell Signaling Technology) were used as primary antibodies to stain patient tumors for overnight incubation at 4°C. CD8 rabbit antimouse antibody (Cat. no. 203035, 1:750; Abcam) and cytokeratin mouse antibody (Cat. no. M3515, 1:500; DAKO) were used. The secondary antibody (prediluted biotinylated horse antimouse/ rabbit IgG secondary antibody) was added for 1-h incubation at room temperature. For signal amplification and detection, the slides were incubated with VECTASTAIN Elite ABC reagent for 30 min and ImmPACT DAB EqV solutions for 30 s. After counterstaining with hematoxylin for 1 min, clearing, and mounting, pictures were taken under a light microscope. The Roche In Situ Cell Death Detection Kit, Fluorescein (Cat. no. 11684795910) was applied for TUNEL assay. Proteinase K solution (Mat No. 1014023; QIAGEN) was used to pretreat the rehydrated sections. Then the TUNEL reaction mixture was incubated on the section in dark at 37°C for 1 h in dark. The section was mounted with VECTASHIELD vibrance antifade mounting medium (Cat. no. H-1800).

# Supplementary Information

# Acknowledgements

This research is funded by the General Research Fund (#17114814 to VWY Lui, Research Grant Council, Hong Kong) and VWY Lui also receives funding from General Research Fund, Research Grant Council, Hong Kong government, Hong Kong SAR (#17121616, #14168517), Research Impact Fund (#R4017-18), the Health and Medical Research Fund (HMRF#15160691, the Health and Medical Research Fund, the Food and Health Bureau, the Government of the Hong Kong Special Administrative Region), University-Industry Collaboration Program (UIM/329; Innovation and Technology Fund, Hong Kong government, Hong Kong SAR), and the Hong Kong Cancer Fund, Hong Kong SAR. Y Liu and W Piao receive funding supports (Postdoctoral Hub PH-ITF Ref.: PiH/052/18 and PiH/234/18 of UIM/329) from the Innovation and Technology Fund, Hong Kong

government. JYK Chan receives funding support by the Dr Stanley Ho Medical Foundation and the General Research Fund (#14109716; #14108818 General Research Fund, Research Grant Council, Hong Kong government, Hong Kong SAR). YX Su receives funding support from Hong Kong Research Grant Council-General Research Fund #17120718. JR Grandis receives funding from National Institudes of Health grants R35CA231998, U54CA209891, R01DE023685, and R01DE028289.

## Authors Contributions

H-L Ngan: conceptualization, data curation, formal analysis, validation, investigation, methodology, and writing—original draft, review, and editing.

Y Liu: conceptualization, data curation, software, formal analysis, validation, investigation, methodology, and writing—original draft, review, and editing.

AY Fong: data curation, formal analysis, validation, and investigation.

PHY Poon: data curation, formal analysis, validation, and investigation.

CK Yeung: data curation, formal analysis, validation, and investigation.

SSM Chan: data curation, formal analysis, supervision, validation, and investigation.

A Lau: data curation, software, formal analysis, validation, investigation, and methodology.

W Piao: data curation, formal analysis, validation, and investigation.

H Li: data curation, formal analysis, validation, and investigation.

JSW Tse: data curation, formal analysis, validation, and investigation.

K-W Lo: writing—review and editing.

SM Chan: data curation, formal analysis, validation, and investigation.

Y-X Su: resources, data curation, and project administration.

JYK Chan: resources, data curation, and project administration.

CW Lau: resources, data curation, and project administration.

GB Mills: writing—review and editing.

JR Grandis: writing—review and editing.

VWY Lui: conceptualization, resources, formal analysis, supervision, funding acquisition, investigation, methodology, project administration, and writing—original draft, review, and editing.

## Conflict of Interest Statement

VWY Lui received a University-Industry Collaboration Program (UIM/329; from the Innovation and Technology Fund, Hong Kong government, and Lee's Pharmaceutical [Hong Kong Limited] in 2018–2020) and served as a scientific consultant for Novartis Pharmaceutical (Hong Kong) Limited (Oct 2015–Oct 2016). JYK Chan served as a consultant for Intuitive Surgical Inc. (Sunnyvale, CA) and advisor for Aptorum Group Ltd. (Hong Kong). JR Grandis is a co-inventor of a cyclic STAT3 decoy and has financial interests in STAT3 Therapeutics, Inc. GB Mills served as a consultant for AstraZeneca, Chrysallis Biotechnology, ImmunoMET, Ionis, Lilly, PDX Pharmaceuticals, Signalchem Lifesciences, Symphogen, Tarveda and Zentalis. GB Mills also has a financial relationship with Catena Pharmaceuticals, ImmunoMet, SignalChem and Tarveda and is holding licensed technologies including HRD assay to Myriad Genetics and DSP patents with Nanostring. Research of GB Mills is sponsored by Nanostring Center of Excellence and Ionis (Provision of tool compounds).

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
