## [Reviewer comments · Life Science Alliance]

Life Science Alliance

MAPK Pathway Mutations in Head and Neck Cancer affect Immune microenvironments and ErbB3 signaling

Hoi-Lam Ngan, Yuchen Liu, Andrew Yuon Fong, Peony Hiu Yan Poon, Chun Kit Yeung, Sharon Suet Man Chan, Alexandria Lau, Wenying Piao, Hui Li, Jessie Sze Wing Tse, Kwok Lo, Sze Man Chan, Yu-Xiong Su, Jason Ying Kuen Chan, Chin Wang Lau, Gordon Mills, Jennifer Grandis, and Vivian Wai Yan Lui

DOI: <https://doi.org/10.26508/lsa.201900545>

Corresponding author(s): *Vivian Wai Yan Lui, The Chinese University of Hong Kong*

Review Timeline:

Submission Date:	2019-09-09
Editorial Decision:	2019-10-15
Revision Received:	2020-03-26
Editorial Decision:	2020-04-14
Revision Received:	2020-04-17
Accepted:	2020-04-17

Scientific Editor: Andrea Leibfried

Transaction Report:

October 15, 2019

Re: Life Science Alliance manuscript #LSA-2019-00545-T

Dr. Vivian Wai Yan Lui
The Chinese University of Hong Kong
School of Biomedical Sciences
Hong Kong

Dear Dr. Lui,

Thank you for submitting your manuscript entitled "MAPK Pathway Mutations in Head and Neck Cancer affect Immune microenvironments and ErbB3 signaling" to Life Science Alliance. The manuscript was assessed by expert reviewers, whose comments are appended to this letter.

As you will see, while the reviewers appreciate your work, they both think that your conclusions are not sufficiently supported by the data provided. They both provide constructive input, however, based on which we would like to invite you to submit a revised version to us should you be able to address the concerns raised. Please note that we will need strong support from the reviewers on the revised version. Importantly, the data shown are not all convincing and improved data need to get provided to better support the main conclusions. Furthermore, the metadata analyses need to be more granular and controlled and justifications / revision of the approaches provided.

Thank you for this interesting contribution to Life Science Alliance. We are looking forward to receiving your revised manuscript.

Sincerely,

B. MANUSCRIPT ORGANIZATION AND FORMATTING:

Reviewer #1 (Comments to the Authors (Required)):

This paper utilised a variety of publicly available data bases (i.e. datasets from the Cancer Genome

Atlas (TCGA), Kaplan-Meier survival curves, CCLE-proteomic, immunotherapy clinical and targeted sequencing data) as well as experimental data to explore the link between activating mutation in components of the MAPK pathway and increased survival of head and neck squamous cell carcinoma (HNSCC) patients. Accordingly, the analysis revealed that increased MAPK signaling correlated with decreased Erb3 phosphorylation, a well-established marker of HNSCC progression, and increased infiltration of CD8+ T-cells in tumours, a marker of high immunoreactive microenvironment.

Overall opinion:

This is an interesting paper which provides evidence that activating mutations in the MAPK pathway, often associated with a "pro-tumorigenic function", constitute good prognostic indicators for HNSCC patients. The approach is scientifically sound. However, there are a number of concerns that must be addressed prior to considering the paper for publication.

Major comments:

Fig. 2d: These data present experimental evidence that increased MAPK signaling suppresses Erb3 phosphorylation. This is critically important to support the idea that increased survival associated with activating mutations in the MAPK pathway could be, at least in part, mediated by decreased Erb3 signalling. However, to be convincing the authors must improve the quality of these data. In particular, some immunoblot images are clearly over-exposed and therefore uninterpretable. The exposure time for each panel must be chosen so that the intensity of the bands in the EGFP extracts is the same between the different conditions of infection (e.g. the level of pErb3 in the EGFP control extract presented next to Ras infected cells should be the same as that presented in EGFP control used for MA2K1 expression. Moreover, the quantification of the chemiluminescent signal should be expressed as ratio, i.e. pErb3/Erb3. The data should represent the mean of at least 3 independent +/- SD or SE, so that the data can be statistically analysed.

Fig. 2e: The authors should explain why pErb3 is detected as a doublet in primary cells. The size of pErb3 does not seem to match across the different cell types. For example, the band corresponding to pErb3 in HSC6 cell lines is lower than in the other extracts. The intensity of the pRSK band in HSC6 cell line is too low to convincingly demonstrate the pharmacological efficacy of GDC-0994. As indicated for panel d, the quantification of the chemiluminescent signal should be expressed as ratio, i.e. pErb3/Erb3. The data should represent the mean of at least 3 independent +/- SD or SE, so that the data can statistically analysed.

Fig. 2g: The quality of the images is very poor. Better images should be provided at a higher magnification so that the intensity and membrane localisation of p-Erb3 can be convincingly compared between MAPK-mutant and -wild HNSCC patient tumors.

Fig. 4d and e: This analysis should include a tumour growth curve to show the effect of activating mutations in MAPK signaling pathway (i.e. mHRAS-G12V and MAPK1-D319N/E320K) on the tumorigenic activity of HNSCC cells. This is essential to correlate the increase in CD8+ T-cell infiltration and cell death caused by MAPK activation with tumor development, according to increased overall survival of MAPK mutant HNSCC patients. Moreover, the time of tumor collection after cell implantation should be clearly indicated. Lastly, the authors must examine the level of p-Erb3 in these tumours to confirm the link between MAPK and Erb3 signaling and further explore the relationship with high immuneactivity of the microenvironment.

Minor comments:

The grammar should be improved throughout the paper and there are many typos that must be corrected.

Reviewer #2 (Comments to the Authors (Required)):

Here, the authors present a molecular landscape that is associated with MAPK-mutant head and neck squamous cell carcinoma (HNSCC). Using mostly TCGA data, they first showed that MAPK-mutant HNSCCs are of a favourable prognostic subtype, regardless of HPV status. To explain this, they showcased two main findings - 1) the association of MAPK-mutant HNSCC with low p-ErbB3 expression, which may result in reduced tumour proliferation; and 2) the presence of a "hot" cytolytic immune landscape, mostly determined by RNA expression. To substantiate their claims, they include some primary MAPK-mutant and MAPK-wt primary HNSCC cell lines.

Overall, the work is well done and rather comprehensive. However, I found it difficult to follow in parts. In particular, it is unclear how the story jumped from ErbB3 to interrogating the immune landscape. On this note, I am also unclear about how the BRAF mutational status mattered to the overall story, let alone explained for the favourable prognosis that is observed with MAPK-mutant HNSCC. In addition to this point, I have the following other points that address specific aspects of the story.

1. Clinical prognostication: I am aware that the granularity of these analyses is limited by the data that is available from the TCGA datasets. However, by presenting the analyses in Figure 1, the authors had overlooked several possible clinical confounders that ought to be considered. Apart from the obvious treatment heterogeneity (Supplementary Fig S1), several clinical parameters such as tumour site (oral cavity vs oropharynx etc), pathological features (including margin and extranodal extension status) were not considered. While it is appreciable that consideration of all these parameters in a multivariable model would be statistically challenging in terms of over-fitting, the authors could explore the "true" prognostic performance of MAPK-mutation status by examining the association in a specific tumour type or at least consider the key factors like margin and extranodal extension status in a multivariable logistic regression analysis.
2. Next, for Figures 1b and 1e, I am unsure what the authors meant when they indicated favourable and unfavourable OS? Is this by a certain cut-off? And if so, what's the justification of using such a cut-off?
3. Association of MAPK-mutated and p-ErbB3: In the same vein, how was the p-ErbB3 defined? Presumably, this was by gene expression and if so, how did the authors normalise for the different RNA raw counts? And did they examine p-ErbB3 or total ErbB3 in Figure 2a?
4. As mentioned above, I am unsure how is the data on the BRAF mutation status relevant for this analysis? The authors have mostly demonstrated the possible interaction between BRAF v600E status and ErbB3 expression; but they have not shown that treatment with a BRAF inhibitor reversed this phenotype? I would recommend additional experiments or perhaps preferred, to streamline the presentation of these results.
5. MAPK-mutation status and the immune landscape: In Figures 4f-i, the authors had not justified their approach of using the arbitrary cut-offs of top 20% and bottom 20% for each of the immune signature scores. Did they perform a sensitivity analysis or was this the cut-off chosen because it yielded the best results in terms of separation of survival curves? What would the curves look like

had they chosen the median or upper most tertile or quartile vs mid-lowest tertiles/quartiles? In the same vein, what were the respective numbers at risk for each subgroup (top and bottom 20%) for each immune score (CYT, T effector, IFNG)? Without these analyses, I am unconvinced by their results, and the authors ought to tone down on their claims.

6. In the same vein, for the heatmap (Figure 4j), colour codes are missing for the respective bars, and likewise, it is not sure how the high vs low subtypes were defined.

7. MAPK-mutation, TMB status and immunotherapy response: I would contest the relevance of TMB status for these analyses. Foremost, the data supporting the concept that TMB predicts response to immunotherapy is controversial and largely based on retrospective data. Next, in line with my earlier comments, what cut-off did the authors use to defined TMB high, since the cut-off varies by tumour types and NGS platforms based on which the molecular profiling data was generated. In fact, TMB is of limited utility in HNSCC, but rather CPS or PDL1 status (KEYNOTE-048 trial) is more predictive of immunotherapy response in HNSCC. On these counts, I will suggest that the authors simplify their analyses and restrict their interpretation to MAPK-mutational status and possibly PD1/PDL1 expression if extraction of that data is possible.

8. I will also suggest that the authors refer to the analyses by Chen YP (Annals of Oncology, 2019; PMID: 30407504), and test the association of the MAPK-status with the immune class described in that paper.

Point-by-Point Reply to Comments:**Reviewer #1:**

We would like to thank the reviewer for the comment that “this is an interesting paper which provides evidence that activating mutations in the MAPK pathway, often associated with a "pro-tumorigenic function", constitute good prognostic indicators for HNSCC patients.” We are grateful for the kind comments of the reviewer which truly helped improving our manuscript. Thank you very much!

Comment 1:

Fig. 2d: These data present experimental evidence that increased MAPK signaling suppresses Erb3 phosphorylation. This is critically important to support the idea that increased survival associated with activating mutations in the MAPK pathway could be, at least in part, mediated by decreased Erb3 signalling. However, to be convincing the authors must improve the quality of these data. In particular, some immunoblot images are clearly over-exposed and therefore uninterpretable. The exposure time for each panel must be chosen so that the intensity of the bands in the EGFP extracts is the same between the different conditions of infection (e.g. the level of pErb3 in the EGFP control extract presented next to Ras infected cells should be the same as that presented in EGFP control used for MA2K1 expression).

Moreover, the quantification of the chemiluminescent signal should be expressed as ratio, i.e. pErb3/Erb3. The data should represent the mean of at least 3 independent +/- SD or SE, so that the data can be statistically analysed.

Our Reply: Thank you very much for the reviewer’s suggestions to further improve the figure quality and presentation, as well as the comment for the statistics. First, we would like to apologize for the use of multiple Western blot sets since a large number of infection sets and thus Western blots were being handled, and for each infection, an EGFP control was included. Thus, across different infection sets, the EGFP bands from individual runs may look different in intensities. We took the reviewer’s comment to make the Western blot presentation more consistent across different independent experimental sets by taking the efforts to repeat again 4 large independent infection sets (N=4 sets total) for all mutants and WTs, and run the gels at the same time for each set of infection to minimize the problem of intensity differences across gene sets (i.e. with one EGFP control across all mutants and WTs). Though one gel can take in 10 samples only, simultaneous running of all samples at the same time in the entire infection set did markedly improve the quality and intensity of bands across different WT and mutants. Please kindly see the **new Fig. 2b** in this revised manuscript. We are so grateful for the reviewer’s suggestions and now all intensities are comparable with the same EGFP control. Care was also taken to avoid overexposures as well.

For p-ErbB3 quantification, **we chose to use “p-ErbB3 normalized to loading” because we followed the RPPA proteomic data from TCPA, with which all protein levels, including p-ErbB3, were normalized to general loading control in an unbiased manner.** We also noted that **total ErbB3 was not correlated with HNSCC patient survival.** Therefore, it would mean that “p-ErbB3 normalized to loading” would be our focus in our Western blot analyses. Yet, we honoured the reviewer’s concern and in this revised manuscript, we performed 4 independent experimental repeats, and we quantified: 1) for p-ErbB3 (normalized to actin for loading; shown in new Figure 2b) and found consistent and significant changes for MAPK pathway mutants in downregulating p-ErbB3 levels normalized to

loading as expected from the TCPA analyses shown by the TCPA bioinformatics. 2) also the p-Erb3/Erb3 ratio per reviewer's suggestion for your kind reference, since some mutants e.g. *MAP2K1* p.K57N are noted to reduce both the levels of p-ErbB3 and total ErbB3, and so the p-ErbB3/ErbB3 ratios for *MAP2K1* WT and mutant may not display statistical difference. Whereas for some mutants, this ratio was still showing statistical significance, such as *ARAF* p.S124F, and *BRAF* p.V600E. The p-Erb3/Erb3 ratio was shown in **Supplementary Figure S4a** for readers' reference due to the limitation of space in the main Figure. Nevertheless, this calculated ratio does provide some degree of reference to total ErbB3 in each mutant set. We thank the reviewer's suggestions to help improving the quality and presentation of our findings. Thank you!

Comment 2:

Fig. 2e: The authors should explain why pErb3 is detected as a doublet in primary cells. The size of pErb3 does not seem to match across the different cell types. For example, the band corresponding to pErb3 in HSC6 cell lines is lower than in the other extracts.

The intensity of the p-RSK band in HSC6 cell line is too low to convincingly demonstrate the pharmacological efficacy of GDC-0994. As indicated for panel d, the quantification of the chemiluminescent signal should be expressed as ratio, i.e. pErb3/Erb3. The data should represent the mean of at least 3 independent +/- SD or SE, so that the data can statistically analysed.

Our Reply: We notice that double bands of p-ErbB3 can happen sometimes when the low percentage SDS-PAGE gel was run for a long time. We apologize for this oversight. We have re-run the Westerns for a shorter time and ensured this was not happening. We also apologize that our gels for different panels were slightly displaced in terms of positioning of the subpanels, thus making the p-ErbB3 or total ErbB3 bands look slightly off in size across the subpanels. We have fixed these problems in the **new Figure 2c** in our revised manuscript.

Concerning reviewer's query on the low level of p-RSK in HSC-6, we hereby confirmed its very low p-RSK level vs. Pt.25 and HSC-4 as shown in the **Review-Supplementary Figure 1**. In fact, HSC-6 has very low endogenous level of p-RSK, which can be seen with either very long exposure (2h), or increased loading amount of protein lysates for this cell line only (100 micrograms of protein lysates for HSC-6 treated with vehicle control, as well as GDC-0994; new Fig. 2c). We hope that this will enable a clearer visual display for the effect of GDC-0994 on p-RSK inhibition in HSC-6. The new figure legend was thus modified accordingly (**p.31 & 32 of the revised manuscript**).

We have repeated the experiment 4 times (N=4 independent repeats). For quantification, as explained above under comment 1, we intended to follow the TCPA RPPA data analysis for all normalization of all protein levels, thus normalizing p-ErbB3 to loading in our Western blots. Nevertheless, we plotted the cumulative plots for both the normalized p-ErbB3 level, as well as the p-ErbB3/ErbB3 ratio in **Supplementary Figure S5b** per reviewer's suggestion. Again, statistical significance was observed consistently for GDC-0994's effect on p-RSK upon ERK inhibition in MAPK-mutant HNSCC models (Pt.25 and HSC-6), as well as in MAPK-WT cells (HSC-4) (N=4 independent experiments), ***which was consistent with the RPPA results of TCPA (p-ErbB3 normalized to loading) as well as our hypothesis that ErbB3 activity is important in MAPK-mutant HNSCC cell***

Review-Supplementary Figure 1

Review-Supplementary Figure 1 - Western blot showing the same molecular size of p-ErbB3 among three different cell lines as well as the low endogenous level of p-RSK in HSC-6.

models. Yet, since the drug GDC-0994 also exerts some effects on total ErbB3 levels in cell line dependent manner with mechanisms unclear (please note the slight downregulation of total ErbB3 in Pt.25, and HSC-4; while slight upregulation in HSC-6), the ratio of p-ErbB3/total ErbB3 would not accurately reflect ErbB3 activity as the denominator values were also changed by the drug, but not in the vehicle control. Nevertheless, the calculated ratios are shown for your kind reference in the new Supplementary Figure 5b.

Comment 3:

Fig. 2g: The quality of the images is very poor. Better images should be provided at a higher magnification so that the intensity and membrane localisation of p-Erb3 can be convincingly compared between MAPK-mutant and -wild HNSCC patient tumors.

Our Reply: We apologize for the poor image quality for some tumors in the original manuscript. We have attempted to re-take pictures for some of the tumors with poor images, and were able to provide better images with high clarity and higher magnification (40 x images) to see the membranous p-ErbB3 staining. Please kindly see the new **Figure 2e** in the revised manuscript. However, because the image quality of the T92 (MAPK-mut) and T90 (MAPK-WT) tumor were still poor, we therefore decided to stain a new pair of MAPK-WT (T75) and MAPK-mut (T61) tumors to replace them. This new pair of tumors are showing consistent results as before that MAPK-mut (T61) has reduced p-ErbB3 staining compared to the MAPK-WT (T75) tumor. Thank you!

Comment 4:

Fig. 4d and e: This analysis should include a tumour growth curve to show the effect of activating mutations in MAPK signaling pathway (i.e. mHRAS-G12V and MAPK1-D319N/E320K) on the tumorigenic activity of HNSCC cells.

This is essential to correlate the increase in CD8+ T-cell infiltration and cell death caused by MAPK activation with tumor development, according to increased overall survival of MAPK mutant HNSCC patients.

Our Reply: We thank the reviewer for the comments. First, we have to deeply apologize that due to the social movements in Hong Kong (from Nov-Dec 2019) and the coronavirus outbreak in Hong Kong (Jan-March 2020, or potentially even much longer), our laboratory have been closed, and with unscheduled intermittent long-term suspensions. It was entirely impossible for us to complete the entire growth curve experiment kindly suggested by the reviewer. Despite all these difficulties, in the month of Jan 2020 prior to the outbreak, we were able to complete a preliminary experiment to test the tumorigenicity of the mHRAS G12S and mMAPK1 D319N mutants in C3H mice (**Review-Supplementary Figure 2**). Consistent with our observed increased apoptosis detected in the mHRAS G12S mutant vs mHRAS-WT, and mMAPK1 D319N mutant vs mMAPK1-WT, we found either a relatively smaller tumor size (mHRAS G12S) or a potential lower rate of tumor formation (mMAPK1 D319N) in MAPK mutant tumors. In this preliminary experiment, a relatively large tumor size of mHRAS-WT tumors than mMAPK1-WT tumors was noted. However, ever after Jan 2020, we were unable to carry out animal experimentation, and

we are so unsure of how long will the coronavirus outbreak last. Yet, we hope that this preliminary piece of data could serve to provide a reasonable answer to the reviewer's concern regarding the tumorigenicity of these mutant tumors.

We also have an additional piece of information regarding the tumorigenicity of MAPK pathway mutant HNSCC tumor, which is interesting to be shared here: with our in-house xenograft established from Pt.25 cells (grown from an HNSCC patient in Hong Kong) which carry both *HRAS* p.G12S and *MAPK1* p.R135K mutations, we found that cells did establish tumors *in vivo* (8/8, 100), but with rather unexpected growth characteristics over time. As shown in **Review-Supplementary Figure 3**, the

Review-Supplementary Figure 3

Review-Supplementary Figure 3 - Tumor growth curves of MAPK pathway mutated Pt.25 primary cell line xenograft as well as the MAPK pathway WT Pt.13 and Pt.16 primary cell line xenografts.

tumors started to grow around day 5 after inoculation, but once the tumors were established, they became very stagnant in growth rate over time and maintained as small size tumors (~40mm³). This is in contrast to the known aggressive nature of HNSCC and in fact, Pt.25 xenografts grew very differently when compared to two other MAPK-WT in-house xenografts from patients' tumor cells we established in Hong Kong (Pt.16, Pt.13; Review-Supplementary Figure 3). Though this is unlikely due to the immune aspects we identified in this study as these human HNSCC xenograft data were performed in nude mice (without CD8+ cells), our findings do suggest likely additional intrinsic properties of MAPK-mutant HNSCC tumors of even more unknown mechanisms other than the ones we could identify by TCGA and TCGA omics analysis (i.e. p-ErbB3 and immune-hot natures). Unfortunately, this represent the limitation of the current omics analyses on TCGA and TCGA data.

In fact, an obvious limitation of the current TCGA and TCGA data is that those tumors are all treatment naïve, which prohibits us to examine potential mechanisms related to good treatment outcomes upon therapy as in TCGA (i.e. treatment-related omics characteristics or related changes in MAPK-mutant tumors in TCGA and TCGA are not available). Lastly, we have previously shown that 2 *MAPK1*-mutants (p.E322K, and recently p.D321N) are particularly sensitivity to EGFR inhibition [1]. These findings may support the notion that MAPK-mutant HNSCC may have other special properties linking to better HNSCC patient outcomes because of their potential responsiveness to certain drug treatments, including EGFR inhibitors. Again, this kind of treatment-related omics data area lacking in the current TCGA and TCGA settings. It is possible that with richer treatment details and omics data linked to drug responses, including tumor shrinkage over time for each patient, more mechanisms can be identified for MAPK-mutant HNSCC tumors.

Comment 5:

Moreover, the time of tumor collection after cell implantation should be clearly indicated. Lastly, the authors must examine the level of p-Erb3 in these tumours to confirm the link between MAPK and Erb3 signaling and further explore the relationship with high immuneactivity of the microenvironment.

Our Reply: We apologize for our oversight. We have detailed the time of tumor collection in the figure legend for Fig. 4d: day 6 for *mMAPK1*-WT vs. *mMAPK1* p.E320K pair due to high levels of tumor cell apoptosis early at day 6 already), and day 11 for *mHRAS*-WT vs. *mHRAS* p.G12V & *mMAPK1*-WT vs. *mMAPK1* p.D319N pairs. Please see **p.33 in the revised manuscript with RED highlight**). Furthermore, we have also performed staining for *mMAPK1*-WT vs *mMAPK1* D319N tumors as well as *mHRAS*-WT vs *mHRAS* p.G12V tumors (grown in C3H mice), and found that both *mHRAS* p.G12V and *mMAPK1* p.D319N mutant tumors resulted in downregulation of p-ErbB3 staining vs. respective WT tumors (**Review-Supplementary Figure 4**). Thus, agreeing with our findings using *in vitro* engineered model (as shown in manuscript **Figure 2b**), and the direct patient tumor staining (as shown in manuscript **Figure 2e**), we further demonstrated in mouse tumor that hotspot mutations of MAPK pathway could lead to p-ErbB3 downregulation *in vivo*.

Review-Supplementary Figure 4 - Decreased expressions of p-ErbB3 (Y1289) were detected in *mHRAS* p.G12V mutant and *mMAPK1* p.D319N mutant (corresponding to MAPK1 p.D321N mutation in human MAPK1) xenografts when compared with their respective mouse WT xenografts by immunohistochemistry on day 11 after tumor inoculation.

As for the link between MAPK-mut/ErbB3 signaling and high immunoreactivity, we apologize that we might have not made it comprehensively clear in our original manuscript (Fig. 4 J) that with a 50% cut-off for immunoreactivity scores of the tumors, there was no significant overlap between low p-ErbB3 RPPA levels (from T CPA) and high immune reactivity scores (i.e. p-ErbB3-low group do not have high IFNG, high CYT, nor high T-eff scores). We have performed the analysis further to cover all cut-offs: ranging from quintile, quartile, tertile as well. The results are shown for your kind reference below (**Review-Supplementary Figure 5**). Across all cut-offs, we found no significant overlaps between low p-ErbB3 RPPA levels (from T CPA) and high immune reactivity scores in HNSCC tumors, strongly indicating the two events are not co-occurring (i.e. p-ErbB3-low group do not have high IFNG, high CYT, nor high T-eff scores).

Review-Supplementary Figure 5 – OncoPrint showing the p-ErbB3 low, IFNG score high, CYT score high and T effectors score high patients defined by 20%, 25%, 33% and 50% cut-off respectively. No significant associations among the p-ErbB3 status and the immune scores were shown by Fisher's exact test.

Review-Supplementary Figure 6

Review-Supplementary Figure 6 - TIMER analysis of p-ErbB3 high vs p-ErbB3 low in TCGA HNSCC cohort with different cut-offs (10%, 20%, 25%, 30%, 33%, 40% and 50% respectively).

Here, for your interest, we also included our TIMER analyses for p-ErbB3 level (high p-ErbB3 and low p-ErbB3, cut-offs at 10% (decile), 20% (quintile), 25% (quartile), 30%, 33% (tertile), 40% and 50% (median)) to examine potential relationship between p-ErbB3 and immunity (**Review-Supplementary Figure 6**). We found that only at certain cut-offs (but not all cut-offs), there appears to have some increases in CD4+T cell population in the patient tumors with low p-ErbB3. This is partially consistent with the tumor model experiment recently put forth by Kumai *et al.* [2].

Comment 6: Minor comments: The grammar should be improved throughout the paper and there are many typos that must be corrected.

Our Reply: We apologize for the careless mistakes and typos made. We have carefully edited the manuscript and have corrected these errors (all in RED highlights in this revised manuscript). Thank you very much.

Reviewer #2:

Comment 1:

Overall, the work is well done and rather comprehensive.

However, I found it difficult to follow in parts. In particular, it is unclear how the story jumped from ErbB3 to interrogating the immune landscape. On this note, I am also unclear about how the BRAF mutational status mattered to the overall story, let alone explained for the favourable prognosis that is observed with MAPK-mutant HNSCC. In addition to this point, I have the following other points that address specific aspects of the story.

Our Reply: Thank you for the very positive and kind comments of the reviewer. Concerning the unexplained jump from ErbB3 to interrogating the immune landscape of MAPK-mutated HNSCC, we truly apologize for our lack of clarity in our presentation in the original manuscript due to word limits. In this revised manuscript, we added back the rationale behind to improve the clarity of our presentation. In addition to our proteomics findings, we further examine, by transcriptomic analysis, if MAPK-mutant HNSCC tumors harbouring immune features favour survival as recent findings in melanoma showed that patients with MAPK pathway mutations have remarkable clinical outcome likely due to increased neo-antigenicity or anti-tumor immune microenvironment [3, 4] (**p.9, highlighted in RED for your kind reference**).

As for the small section on *BRAF* mutational status, we agree with the reviewer to better streamline the result presentation. We have addressed this under **Our Reply to Comment 4** below. Thank you very much!

Comment 2:

Clinical prognostication: I am aware that the granularity of these analyses is limited by the data that is available from the TCGA datasets. However, by presenting the analyses in Figure 1, the authors had overlooked several possible clinical confounders that ought to be considered. Apart from the obvious treatment heterogeneity (Supplementary Fig S1), several clinical parameters such as tumour site (oral cavity vs oropharynx etc), pathological features (including margin and extranodal extension status) were not considered. While it is appreciable that consideration of all these parameters in a multivariable model would be statistically challenging in terms of over-fitting, the authors could explore the "true" prognostic performance of MAPK-mutation status by examining the association in a specific tumour type or at least consider the key factors like margin and extranodal extension status in a multivariable logistic regression analysis.

Our Reply: Thank you for the reviewer's expert suggestions. We have performed the multivariable logistic regression on the dataset and the detailed results are shown in the **Review-Supplementary Table 1** below. Among the predictors (MAPK mutation status, tumor primary site, tumor margin status and extranodal extension status), only MAPK mutation and extranodal extension are significantly associated with patient outcome (patient death events). The coefficient estimate of the variable MAPK mutation is -0.57104, which is negative. It means an increase in MAPK mutation will be associated with a decreased probability of patient death, which is consistent with our hypothesis.

We further examined if MAPK mutation and extranodal extension are associated with each other by Fisher's exact test and showed that there is no significant association between these two factors (**Review-Supplementary Table 2**), only a trend to be associated with each other). Therefore, we think

that MAPK pathway mutation can be one of the strong prognostic biomarkers indicating better outcome in HNSCC.

Review-Supplementary Table 1

	Estimate	Std. Error	z value	Pr(> z)
(Intercept)	-0.439169402	0.613327627	-0.71604	0.473964
MAPK_MUT	-0.5710425	0.258661157	-2.20769	0.027266
PRIMARY_SITE_Base of tongue	-0.166506288	0.683996536	-0.24343	0.807671
PRIMARY_SITE_Buccal Mucosa	0.142160154	0.693171595	0.205087	0.837505
PRIMARY_SITE_Floor of mouth	0.517205711	0.579667153	0.892246	0.372261
PRIMARY_SITE_Hard Palate	-0.053560099	0.999958174	-0.05356	0.957284
PRIMARY_SITE_Hypopharynx	0.036091438	0.838058663	0.043066	0.965649
PRIMARY_SITE_Larynx	0.213472101	0.55232921	0.386494	0.699131
PRIMARY_SITE_Lip	-0.260889489	1.376620221	-0.18951	0.84969
PRIMARY_SITE_Oral Cavity	0.797976449	0.573813109	1.390656	0.16433
PRIMARY_SITE_Oral Tongue	0.178563675	0.545593405	0.327283	0.743454
PRIMARY_SITE_Oropharynx	-0.579721992	0.908209781	-0.63831	0.52327
PRIMARY_SITE_Tonsil	-1.037456335	0.643788228	-1.61149	0.107074
PATH_MARGIN_Close	0.160317938	0.468654925	0.342081	0.73229
PATH_MARGIN_Negative	0.135365443	0.360431234	0.375565	0.70724
PATH_MARGIN_Positive	0.870619308	0.446450515	1.950091	0.051165
EXTRACAPSULAR_SPREAD_PATHOLOGIC_Gross Extension	0.730486072	0.408431153	1.788517	0.073693
EXTRACAPSULAR_SPREAD_PATHOLOGIC_Microscopic Extension	0.574629331	0.310655693	1.849731	0.064352
EXTRACAPSULAR_SPREAD_PATHOLOGIC_No Extranodal Extension	-0.559985287	0.240829723	-2.32523	0.020059

Review-Supplementary Table 1 – Multivariable logistic regression examining the association between the clinical features of HNSCC patients and their outcome.

Review-Supplementary Table 2

	MAPK pathway Mutated (n=95)		MAPK pathway WT (n=415)		χ ² P-value	Fisher's P-value (2-sided)
Extracapsular spread pathologic						
Gross Extension	6	6.3%	31	7.5%	0.064	0.05936
Microscopic Extension	7	7.4%	70	16.9%		
No Extranodal Extension	50	52.6%	191	46.0%		
N.A.	32	33.7%	123	29.6%		

Review-Supplementary Table 2 – Fisher's exact test showing no significant association between MAPK pathway mutation and Extracapsular spread (extranodal extension) status. (Clinical data from TCGA HNSCC n=512 cohort)

Comment 2:

Next, for Figures 1b and 1e, I am unsure what the authors meant when they indicated favourable and unfavourable OS? Is this by a certain cut-off? And if so, what's the justification of using such a cut-off?

Our Reply: The favourable/unfavourable OS indicated in Figure 1b & 1e are defined by log-rank P-value (P<0.05) generated from Kaplan-Meier curves as in Figures 1c and 1d. Patients are stratified by the specified status, e.g. MAPK1 pathway mut vs. MAPK1 pathway WT, PI3K pathway mut vs. PI3K pathway WT, HPV(+) vs. HPV(-), etc. Then, the OS advantages were analyzed by Kaplan-Meier survival analyses. Only those patient groups with log-rank P-value <0.05 (significant) would we consider them having favourable/unfavourable OS. Hope this clarifies our definition. We have therefore included a more detailed clarification in the methodology section on **p.18 of the revised manuscript** in RED highlights.

Comment 3:

Association of MAPK-mutated and p-ErbB3: In the same vein, how was the p-ErbB3 defined? Presumably, this was by gene expression and if so, how did the authors normalised for the different RNA raw counts? And did they examine p-ErbB3 or total ErbB3 in Figure 2a?

Our Reply: Thank you. In the manuscript, p-ErbB3 (i.e. phospho-ErbB3 protein level) specifically refers to the normalized RPPA phospho-ErbB3 (Y1289) level from TCPA RPPA Level 4 proteomic data. Per TCPA RPPA information released, the loading is normalized to pan-protein expression levels in the array ([https://bioinformatics.mdanderson.org/public-software/tcpa/?_ga=2.147687607.186219550.](https://bioinformatics.mdanderson.org/public-software/tcpa/?_ga=2.147687607.186219550.1584258029-509834093.1584258029)

1584258029-509834093.1584258029). We extracted the level 4 normalized p-ErbB3 protein expression for HNSCC tumors from TCPA database. The p-ErbB3 high/low is defined by the median cut-off of the p-ErbB3 RPPA value, which is the same criteria adopted by TCPA for survival analyses/plots in TCPA website (https://tcpaportal.org/tcpa/survival_analysis.html). We have modified our figure legend by indicating “median cut-off” (**p.31 in RED highlights**). TCPA-survival

Review-Supplementary Figure 7

Review-Supplementary Figure 7 – Kaplan-Meier overall survival curves for total ErbB3 high patients vs total ErbB3 low patients (TCPA HNSCC cohort, median cut-off).

analysis indicates that low p-ErbB3 (Y1289) (median cut-off; Log-rank P=0.0006 in Figure 2a of our manuscript) is associated with good patient outcomes, but not total ErbB3 levels (median cut-off; Log-rank P value=0.84285, n.s.). We have extracted the figure from TCPA-survival analysis for total ErbB3 level for your kind reference below (**Review-Supplementary Figure 7**).

Comment 4:

As mentioned above, I am unsure how is the data on the BRAF mutation status relevant for this analysis? The authors have mostly demonstrated the possible interaction between BRAF v600E status and ErbB3 expression; but they have not shown that treatment with a BRAF inhibitor reversed this phenotype? I would recommend additional experiments or perhaps preferred, to streamline the presentation of these results.

Our Reply: We agree with the reviewer that the BRAF result on inverse correlation with p-ErbB3 appear to be rather irrelevant. Thus, we have removed this small part, which really help better streamlining the presentation of MAPK pathway mutants' effects on p-ErbB3 in HNSCC. Thank you so much!

Comment 5:

MAPK-mutation status and the immune landscape: In Figures 4f-i, the authors had not justified their approach of using the arbitrary cut-offs of top 20% and bottom 20% for each of the immune signature scores. Did they perform a sensitivity analysis or was this the cut-off chosen because it yielded the

best results in terms of separation of survival curves? What would the curves look like had they chosen the median or upper most tertile or quartile vs mid-lowest tertiles/quartiles? In the same vein, what were the respective numbers at risk for each subgroup (top and bottom 20%) for each immune score (CYT, T effector, IFNG)? Without these analyses, I am unconvinced by their results, and the authors ought to tone down on their claims.

Our Reply: Thank you for your kind comments. To clarify, our original cut-off of 20% was an arbitrary cut-off. Per reviewer's requested, we have then analysed the data again with quintile (20%), quartile (25%), tertile (33%) and median (50%) cut-offs, with the number at risk shown below each graph for all three immune scores (CYT, IFNG, T-eff scores) (**Review-Supplementary Figure 8, which is also shown in the revised manuscript as Supplementary Figure S11**). For IFNG score, across all cut-offs, the survival associations all reached statistical significance of $P < 0.05$. For CYT score, the survival associations reached statistical significance of $P < 0.05$ for all cut-offs except for the most relaxed cut-off of 50% ($P = 0.2$). For T-eff score, the survival associations reached statistical significance of $P < 0.05$ for all cut-offs except for a near significant trend of $P = 0.083$ in quartile cut-off. Though these findings are largely similar to our original findings based on the arbitrary 20% cut-off, we decide to modify the text accordingly **on p.13 of the revised manuscript in RED highlight**.

Review-Supplementary Figure 8

Quintile (20%) Cut-off

Quartile (25%) Cut-off

Tertile (33%) Cut-off

Median (50%) Cut-off

Review-Supplementary Figure 8 – Kaplan-Meier overall survival curves for immune score high patients vs immune score low patients (CYT score, IFNG score and T effector score respectively, with 20%, 25%, 33% and 50% cut-off).

Comment 6:

In the same vein, for the heatmap (Figure 4j), colour codes are missing for the respective bars, and likewise, it is not sure how the high vs low subtypes were defined.

Our Reply: We apologize for the missing labelling for the color coding for Figure 4j. We have added back the detailed color coding information in the legend of the figure in this revised manuscript (**p.34 in RED highlights**). Furthermore, in the original manuscript, we have used the median cut-off (50%

Review-Supplementary Figure 5

Review-Supplementary Figure 5 – Oncoprint showing the p-ErbB3 low, IFNG score high, CYT score high and T effectors score high patients defined by 20%, 25%, 33% and 50% cut-off respectively. No significant associations among the p-ErbB3 status and the immune scores were shown by Fisher's exact test.

cut-off) to define all three immune scores and the p-ErbB3 level (just to be consistent with the p-ErbB3 cut-off defined by TCGA-RPPA dataset, and in our Figure 2A). Here, we have performed all cut-offs to examine their relationship more vigorously and found that across all cut-offs examined (quintile, quartile, tertile, and median cut-offs), MAPK pathway mutations are not overlapping with the immune-high scores (**Review-Supplementary Figure 5**). For the comprehensiveness of reporting and the relatedness with **Figure 4f-h** with quintile cut-off (20%), we have in this revised manuscript, added also the 20% cut-offs for the readers' information in the **new figure 4j**. Results on the remaining cut-offs are shown in **Supplementary Figure S11**.

Comment 7:

MAPK-mutation, TMB status and immunotherapy response: I would contest the relevance of TMB status for these analyses. Foremost, the data supporting the concept that TMB predicts response to immunotherapy is controversial and largely based on retrospective data. Next, in line with my earlier comments, what cut-off did the authors use to defined TMB high, since the cut-off varies by tumour types and NGS platforms based on which the molecular profiling data was generated. In fact, TMB is of limited utility in HNSCC, but rather CPS or PDL1 status (KEYNOTE-048 trial) is more predictive of immunotherapy response in HNSCC. On these counts, I will suggest that the authors simplify their analyses and restrict their interpretation to MAPK-mutational status and possibly PD1/PDL1 expression if extraction of that data is possible.

Our Reply: Thank you for the reviewer's expert comment. In this study, the TMB cut-off value was directly adopted from the original publication that (10.3) for head and neck cancer, the TMB cut-off was 10.3 (calculated as the number of non-synonymous somatic mutations, single nucleotide variants and small insertions/deletions, per mega-base in coding regions as defined by the authors: (<https://www.ncbi.nlm.nih.gov/pubmed/30643254>) [5]. We employed the same $TMB \geq 10.3$ as TMB-high per original publication. We have modified the figure legend for such definitions of TMB-high vs TMB-low, please kindly see **p.34 for the revised legend**.

We agree with the reviewer that in HNSCC, TMB is of limited utility as recently recognized in the field. This seems to be consistent with our findings in the original Figures 5f and 5h that in HNSCC, TMB was not significantly associated with improved patient outcomes. Because of this particular concern, we have modified our text accordingly in the revised manuscript “ Whereas TMB-high status only demonstrated a trend for better outcome (Supplementary Fig.S12a), consistent with recent clinical findings that in HNSCC, TMB status may not accurately predict patient outcome (as compared to PD-L1 status) [6]” (**p.14 in RED highlights**). The original TMB figures for HNSCC are thus moved to **Supplementary Figure 12a & 12b**.

As for the PD1/PDL1 expression data, since it is not available from the publication, we could not perform such analysis.

Comment 8:

I will also suggest that the authors refer to the analyses by Chen YP (Annals of Oncology, 2019; PMID: 30407504), and test the association of the MAPK-status with the immune class described in that paper.

Our Reply: Thanks a lot for the expert suggestion. We have shown in Figure 5e that in oral cancer that MAPK-mut patients have better OS with ICI. We thank the reviewer for suggesting us to analyze the HNCC-MAPK status using the immune class defined by Chen YP (Annals of Oncology, 2019; PMID: 30407504). Our result showed that with this immune class definition, MAPK-mut status is significantly associated with “Immune class”, P=0.0195 (Fisher’s exact test) with an Odds Ratio of 0.5313 in head and neck oral cavity dataset (N=309). We have thus included this new finding into our revised manuscript (**new manuscript Fig.5f and p.14 with RED highlights**).

References:

1. Van Allen, E.M., et al., *Genomic Correlate of Exceptional Erlotinib Response in Head and Neck Squamous Cell Carcinoma*. JAMA oncology, 2015. **1**(2): p. 238-44.
2. Kumai, T., et al., *Targeting HER-3 to elicit antitumor helper T cells against head and neck squamous cell carcinoma*. Scientific Reports, 2015. **5**.
3. Cadley, J., et al., *Mutation burden in conjunction with MAPK-pathway mutation status as a prognostic biomarker of overall melanoma survival*. Journal of Clinical Oncology, 2018. **36**(15).
4. Veatch, J.R., et al., *Tumor-infiltrating BRAFV600E-specific CD4+ T cells correlated with complete clinical response in melanoma*. J Clin Invest, 2018. **128**(4): p. 1563-1568.
5. Samstein, R.M., et al., *Tumor mutational load predicts survival after immunotherapy across multiple cancer types*. Nat Genet, 2019. **51**(2): p. 202-206.
6. Cohen, E.E.W., et al., *The Society for Immunotherapy of Cancer consensus statement on immunotherapy for the treatment of squamous cell carcinoma of the head and neck (HNSCC)*. J Immunother Cancer, 2019. **7**(1): p. 184.

April 14, 2020

RE: Life Science Alliance Manuscript #LSA-2019-00545-TR

Dr. Vivian Wai Yan Lui
The Chinese University of Hong Kong
School of Biomedical Sciences

Dear Dr. Lui,

Thank you for submitting your revised manuscript entitled "MAPK Pathway Mutations in Head and Neck Cancer affect Immune microenvironments and ErbB3 signaling". As you will see, the reviewers appreciate the introduced changes, and we would thus be happy to publish your paper in Life Science Alliance pending final revisions:

- please address the remaining reviewer concerns
- some of the figures are too busy and labels cannot be read at normal figure size; this needs to get addressed - you can introduce additional figures to do so. Please also see our author guidelines regarding figure preparation.
- please upload the supplementary figures as individual, single-page files;
- Fig S10 and S11 span two pages and labels cannot be read; please revise - you can introduce further S figures
- please add callouts to either all panels for Fig S1, or mention only S1 in all instances - currently only some panels are called-out in the ms text
- you have a callout to table S6 - please fix; the current table S5 is also labeled as S6 on its second page
- the figure legend to figure S11 mentions a panel a); please fix
- please provide all tables in docx or excel format
- please be more specific for the RNA-seq data downloaded and used (add accession codes and references)
- the manuscript must include a statement in the Materials and Methods identifying the institutional and/or licensing committee approving the animal experiments

A. FINAL FILES:

B. MANUSCRIPT ORGANIZATION AND FORMATTING:

Sincerely,

Reviewer #1 (Comments to the Authors (Required)):

The authors have satisfactorily addressed the points raised in my initial reviewed.

Reviewer #2 (Comments to the Authors (Required)):

The authors have adequately addressed my concerns. The manuscript reads a lot better now. I have the following minor points below.

1. I would suggest the authors include the sample size and number of events for their multivariable analyses on prognosis for MAPK. From the additional analysis they provided, it is difficult to judge if the model is over-fitted without knowledge on the number of events.
2. There are some inconsistencies with typos - for example "Dendritic" in upper case but other cell types were spelt in lower case.

Overall, a well done piece of work that advances our understanding of MAPK mutant Head neck cancers.

Point-by-Point Reply to Comments:**Reviewer #1 (Comments to the Authors (Required)):****Comment 1:**

The authors have satisfactorily addressed the points raised in my initial reviewed.

Our Reply: We would like to thank for the reviewer's positive comment.

Reviewer #2 (Comments to the Authors (Required)):**Comment 1:**

The authors have adequately addressed my concerns. The manuscript reads a lot better now. I have the following minor points below.

Our Reply: We would like to thank for the advices and comments suggested by the reviewer to improve the manuscript.

Comment 2:

1. I would suggest the authors include the sample size and number of events for their multivariable analyses on prognosis for MAPK. From the additional analysis they provided, it is difficult to judge if the model is over-fitted without knowledge on the number of events.

Our Reply: Thank you for the suggestion from the reviewer. We have included the sample size and number of events in the new multivariable analyses as shown in Review-supplementary table 1 below.

Review-Supplementary Table 1

	N number	Estimate	Std. Error	z value	Pr(> z)
(Intercept)		-0.4391694	0.6133276	-0.7160437	0.473964
MAPK_MUT	95	-0.5710425	0.2586612	-2.2076856	0.027266
PRIMARY_SITE_Base of tongue	24	-0.1665063	0.6839965	-0.2434315	0.807671
PRIMARY_SITE_Buccal Mucosa	23	0.1421602	0.6931716	0.2050865	0.837505
PRIMARY_SITE_Floor of mouth	62	0.5172057	0.5796672	0.8922460	0.372261
PRIMARY_SITE_Hard Palate	7	-0.0535601	0.9999582	-0.0535623	0.957284
PRIMARY_SITE_Hypopharynx	10	0.0360914	0.8380587	0.0430655	0.965649
PRIMARY_SITE_Larynx	111	0.2134721	0.5523292	0.3864943	0.699131
PRIMARY_SITE_Lip	3	-0.2608895	1.3766202	-0.1895145	0.849690
PRIMARY_SITE_Oral Cavity	70	0.7979764	0.5738131	1.3906557	0.164330
PRIMARY_SITE_Oral Tongue	130	0.1785637	0.5455934	0.3272834	0.743454
PRIMARY_SITE_Oropharynx	9	-0.5797220	0.9082098	-0.6383129	0.523270
PRIMARY_SITE_Tonsil	43	-1.0374563	0.6437882	-1.6114870	0.107074
PATH_MARGIN_Close	50	0.1603179	0.4686549	0.3420810	0.732290
PATH_MARGIN_Negative	350	0.1353654	0.3604312	0.3755652	0.707240
PATH_MARGIN_Positive	57	0.8706193	0.4464505	1.9500914	0.051165
EXTRACAPSULAR_SPREAD_PATHOLOGIC_Gross Extension	37	0.7304861	0.4084312	1.7885170	0.073693
EXTRACAPSULAR_SPREAD_PATHOLOGIC_Microscopic Extension	77	0.5746293	0.3106557	1.8497306	0.064352
EXTRACAPSULAR_SPREAD_PATHOLOGIC_No Extranodal Extension	241	-0.5599853	0.2408297	-2.3252333	0.020059

Review-Supplementary Table 1 – Multivariable logistic regression examining the association between the clinical features of HNSCC patients and their outcome.

Comment 3:

2. There are some inconsistencies with typos - for example "Dendritic" in upper case but other cell types were spelt in lower case.

Our Reply: Thank you for the reminder from the reviewer. We have amended these typos throughout the manuscript.

Comment 4:

Overall, a well done piece of work that advances our understanding of MAPK mutant Head neck cancers.

Our Reply: We would like to thank for the reviewer's positive comment.

April 17, 2020

RE: Life Science Alliance Manuscript #LSA-2019-00545-TRR

Dr. Vivian Wai Yan Lui
The Chinese University of Hong Kong
School of Biomedical Sciences

Dear Dr. Lui,

Thank you for submitting your Research Article entitled "MAPK Pathway Mutations in Head and Neck Cancer affect Immune microenvironments and ErbB3 signaling". I appreciate the introduced changes and the information provided in the point-by-point response, and it is a pleasure to let you know that your manuscript is now accepted for publication in Life Science Alliance. Congratulations on this interesting work.

DISTRIBUTION OF MATERIALS:

Again, congratulations on a very nice paper. I hope you found the review process to be constructive and are pleased with how the manuscript was handled editorially. We look forward to future exciting submissions from your lab.

Sincerely,
